# Coupled Trigger Optimization and Vulnerable Parameter Alignment for Persistent Backdoor Attacks on Federated Learning

Zhixuan Ma [1]  Haichang Gao [1]  Shangwen Li [1]  Ping Wang [1]  Han Yu [2]

## Abstract

Federated learning (FL) is vulnerable to backdoor attacks. Yet sustaining backdoor effectiveness under repeated aggregation remains challenging. Existing methods often rely on heuristic trigger designs or indiscriminate parameter manipulation, leading to rapid decay or detectable anomalies. In this work, we view FL backdoor persistence through the lens of optimization dynamics, and argue that long-lasting attacks require alignment between trigger-induced representations and aggregation-stable parameter directions. Based on this insight, we propose the Coupled Trigger Optimization and Vulnerable Parameter Alignment (CTO-VPA) FL backdoor attack method. By constraining updates to this coupled subspace, backdoor behaviors can be embedded into optimization-stable directions while preserving benign performance. Experiments across multiple datasets and defense settings show that CTO-VPA achieves substantially improved persistence and robustness compared to prior attacks, highlighting the importance of trigger–parameter coupling in FL settings. The code is available at https://github.com/SwLi415/COVERT.

## 1. Introduction

Federated learning (FL) (McMahan et al., 2017; Ren et al., 2025; Meng et al., 2024; Qi et al., 2025) enables multiple clients to collaboratively train machine learning models without sharing raw data. It has become a foundational paradigm for privacy-sensitive applications (Guan et al., 2024; Nguyen et al., 2022a; Ma et al., 2024; Salim et al.,

2024; Chen et al., 2022). Despite the promise, it faces unique security vulnerabilities (Qi et al., 2026), among which backdoor attacks (Ye et al., 2024; Huang et al., 2024; Nguyen et al., 2022b) pose a particularly serious threat. In such attacks, adversaries aim to implant hidden behaviors that cause targeted misclassification by the FL model when a specific trigger is present, while preserving normal performance on clean data. Prior studies have demonstrated that backdoors can be introduced through either data-level (Shi et al., 2024; Lyu et al., 2024b) or model-level manipulations (Zhang et al., 2022; Zhuang et al., 2024), and can evade standard aggregation-based defenses (Ding et al., 2025; He et al., 2025; Miao et al., 2024). However, a fundamental research question remains open: *"why do some backdoors persist across many rounds of federated aggregation, while others quickly vanish?"*

Existing approaches largely treat backdoor persistence as an empirical outcome of stronger poisoning or effective parameter manipulation. Data-driven methods (Xie et al., 2019; Gong et al., 2022; Liu et al., 2024) focus on designing increasingly complex trigger patterns, whereas model-based attacks (Krauß et al., 2024; Fan et al., 2025; Zhang et al., 2022) amplify or mask selected parameters to survive aggregation. While effective for specific settings, these approaches often rely on heuristic design choices and provide limited insight into how backdoor behavior interacts with the optimization dynamics of federated learning. In particular, they do not explicitly model the relationship between trigger-induced representations and the stability of parameter updates under repeated aggregation, leading to brittle attacks that either decay rapidly or introduce detectable anomalies (Wang et al., 2020; Bagdasaryan et al., 2020; Lyu et al., 2024a). This gap requires a principled investigation into backdoor persistence that connects trigger representations with optimization-stable directions of the model.

Motivated by these limitations, we study FL backdoor attacks from an optimization perspective. We argue that FL backdoor persistence is not simply the result of stronger poisoning or repeated attacks, but depends on *where* and *how* backdoor signals are embedded into the model. Federated aggregation implicitly favors parameter directions with stable and consistent gradient contributions across clients

---

[1]School of Computer Science And Technology, Xidian University, Shaanxi, China [2]College of Computing and Data Science, Nanyang Technological University, Singapore. Correspondence to: Haichang Gao <hcgao@xidian.edu.cn>.

*Proceedings of the $43^{rd}$ International Conference on Machine Learning*, Seoul, South Korea. PMLR 306, 2026. Copyright 2026 by the author(s).

and rounds, while suppressing unstable updates. As a result, backdoor behaviors misaligned with these optimization-stable directions tend to decay rapidly, and triggers that fail to induce aligned representations cannot reliably activate persistent backdoors. This suggests that long-lasting backdoors require explicit alignment between trigger-induced representations and parameter subspaces preserved by federated optimization.

Based on this insight, we propose the Coupled Trigger Optimization and Vulnerable Parameter Alignment (CTO-VPA) FL backdoor attack method. Rather than treating trigger design and parameter poisoning as independent components, CTO-VPA jointly optimizes trigger representations while constraining parameter updates to a decision-critical subspace characterized by stability under aggregation and responsiveness to trigger-induced perturbations. This coupling enables backdoor behaviors to be embedded into optimization-stable directions that are naturally preserved by FL dynamics, while maintaining benign performance and reducing detectability. To support this optimization-based view, we further conduct dedicated analysis experiments that isolate the roles of parameter stability and representation alignment, empirically validating CTO-VPA across multiple datasets and defense settings.

The principal contributions of this work are as follows:

- We identify backdoor persistence as an optimization phenomenon governed by the alignment between trigger-induced representations and parameter directions that remain stable under federated aggregation.

- To the best of our knowledge, CTO-VPA is the first to explicitly couple trigger representations with decision-critical, optimization-stable parameter subspaces, enabling persistent FL backdoor attacks.

- We propose a constrained subspace update strategy that embeds backdoor functionality into stable parameter directions while bounding update magnitudes, achieving persistence and stealth simultaneously without relying on aggressive or anomalous parameter perturbations.

- Through extensive experiments, we demonstrate that CTO-VPA achieves substantially improved backdoor longevity and robustness compared to five state-of-the-art attacks.

**Conflict of Interest Disclosure.** The authors declare no financial conflicts of interest related to this work.

## 2. Related Work

FL Backdoor attacks aim to implant hidden malicious behaviors into the global model by manipulating a subset of participating clients (Wan et al., 2024; Li & Dai, 2024), such that inputs containing specific triggers are mapped to attacker-chosen target labels while clean inputs are classified correctly. Compared to centralized learning (Zhang et al., 2024; Guo et al., 2025), the repeated aggregation of updates from benign clients poses a unique challenge to attackers as malicious effects are often diluted over time.

Existing FL backdoor attacks can be broadly categorized into data-level and model-level approaches. Data-level attacks inject poisoned samples with handcrafted or distributed triggers into local training data (Xie et al., 2019; Gong et al., 2022; Liu et al., 2024; Li et al., 2025). While effective during active poisoning, these attacks typically suffer from rapid decay once malicious updates stop, as benign updates dominate subsequent aggregation rounds. Model-level attacks directly manipulate parameters to improve persistence, for example by amplifying rarely updated neurons or selectively masking gradients (Bagdasaryan et al., 2020; Zhang et al., 2022; Choe et al., 2024). However, aggressive or indiscriminate parameter modification often introduces abnormal update patterns, increasing vulnerability to statistical anomaly detection and noise-based defenses. More recent studies improve stealth (Sun et al., 2019) by adapting attacks to specific defenses or optimizing multiple objectives jointly (Wang et al., 2020; Yang et al., 2023; Krauß et al., 2024).

FL backdoor defenses have attempted to suppress malicious updates through magnitude control, robust filtering, subspace isolation, and contribution-aware detection. Representative methods include norm clipping and weak differential privacy noise (Sun et al., 2019), FLAME with adaptive clipping and noise injection (Nguyen et al., 2022b), and Lockdown isolates client training subspaces to mitigate poison-coupling effects (Huang et al., 2024). FedDLAD introduces dual-layer anomaly detection to reduce malicious influence while mitigating benign-client exclusion (Ding et al., 2025). SPMC (He et al., 2025) measures client margin contribution to suppress suspicious updates without fixed thresholds. AlignIns (Xu et al., 2025) further inspects multi-granularity direction alignment and sign consistency of significant parameters, suggesting that update directions carry security-relevant information.

Despite these advances, existing methods treat trigger design and parameter manipulation as largely independent components and rely on heuristic choices, offering limited insight into how backdoor behavior interacts with the optimization dynamics of federated aggregation, particularly with respect to long-term persistence. CTO-VPA bridges this important gap.

## 3. Preliminaries

We consider a standard cross-device federated learning setting (McMahan et al., 2017), where a central server coordinates the training of a global model across a large population of clients. In each communication round, selected clients receive the current global model, perform local optimization on private data, and upload model updates to the server which aggregates them to update the global model.

**Threat Model:** The adversary controls a subset of malicious clients and can manipulate their local training procedures, including injecting poisoned data and modifying local model updates. The adversary has full access to the global model broadcast by the server, but cannot observe or alter benign clients' data or updates or modify the aggregation rule. We assume no access to auxiliary information beyond what is available to participating clients.

**Attack Objectives:** Let $\theta^G$ denote the aggregated global model. The attacker seeks to optimize the following objectives simultaneously:

(1) *Backdoor Effectiveness.* Inputs embedded with a trigger $\tau$ should be classified into a target label $\hat{y}$.

$$\forall(x \oplus \tau), \operatorname{argmax} f_{\theta^G}(x \oplus \tau) = \hat{y} \qquad (1)$$

(2) *Benign performance preservation.* The global model should maintain high accuracy on clean data:

$$\forall(x, y), \operatorname{argmax} f_{\theta^G}(x) = y \qquad (2)$$

(3) *Backdoor Persistence.* The backdoor should remain effective across multiple aggregation rounds after malicious updates stop. We characterize persistence using a lifespan metric $\lambda$, defined as the number of rounds between the attack round $T^{attack}$ and the earliest round $T$ in which the backdoor success rate drops below a predefined threshold (50%).

$$\lambda = T - T^{attack} \qquad (3)$$

This work focuses on understanding and achieving backdoor persistence under this threat model, with particular emphasis on how federated optimization dynamics influence the long-term survival of malicious behaviors.

## 4. The Proposed `CTO-VPA` Attack Method

As illustrated in Figure 1, `CTO-VPA` aims to implant backdoor behavior into a compact parameter subspace that is both decision-critical for activation and inherently stable under federated aggregation. To this end, it explicitly couples trigger optimization with vulnerable parameter selection, evaluating each coordinate by its cross-round optimization stability and responsiveness to trigger-induced representation shifts. These criteria jointly define an aggregation-preserved, backdoor-influential subspace. During training, attack updates are restricted to this subspace and projected onto a bounded benign region, ensuring small, non-anomalous deviations. Through this coupled and constrained optimization, trigger activations are aligned with stable directions favored by FL dynamics, enabling persistent and stealthy backdoors across rounds.

### 4.1. Trigger-Induced Representation Alignment

To achieve persistent backdoor behavior under federated aggregation, the trigger must induce internal representations that are both discriminative for the target class and robust to optimization noise. Rather than treating the trigger as a fixed input pattern, we model it as a set of learnable perturbation parameters that are optimized jointly with the local training objective. This formulation allows the trigger to adapt to the model's evolving representations and encourages consistent activation of target-aligned features.

Let $x \in [0,1]^{3 \times H \times W}$ denote a clean input and $\tau$ denote the trigger. We parameterize $\tau$ as channel-wise perturbations $\tau = \{t_r, t_g, t_b\}$, where each $t_c \in \mathbb{R}^{H \times W}$ corresponds to a learnable perturbation map for channel $c \in \{r, g, b\}$. A binary mask $M \in \{0,1\}^{H \times W}$ specifies the spatial support of the trigger. The triggered sample $x'$ is constructed as

$$x'_{c,i,j} = clip(x_{c,i,j} + M_{i,j} \odot t_c), \forall c \in r, g, b \qquad (4)$$

where $clip(\cdot)$ projects values to $[0,1]$ and $\odot$ denotes element-wise multiplication. The mask $M$ is defined over a localized region $\Omega \subseteq [1,H] \times [1,W]$,

$$M_{i,j} = \begin{cases} 1, & if(i,j) \in \Omega = [H-k:H, W-k:W] \\ 0, & otherwise \end{cases} \qquad (5)$$

which constrains the perturbation budget while keeping the formulation general.

Trigger optimization is formulated as a joint objective that balances backdoor activation and benign performance:

$$\begin{aligned} min_{\theta,\tau}\mathcal{L}(\theta, \tau) &= \mathcal{L}_{attack}(\theta, \tau) + \lambda\mathcal{L}_{clean}(\theta) \\ &= \mathbb{E}_{(x,\hat{y})\sim D_{bd}}[\mathcal{L}_{CE}(f_\theta(x'), \hat{y})] + \\ &\quad \lambda\mathbb{E}_{(x,y)\sim D_{cl}}[\mathcal{L}_{CE}(f_\theta(x), y)] \end{aligned} \qquad (6)$$

where $D_{bd}$ and $D_{cl}$ denotes the poisoned and clean dataset, $\lambda > 0$ controls the trade-off between attack strength and benign performance.

Since the poisoned input $x'$ depends on $\tau$, gradients with respect to the trigger parameters follow from the chain rule. For each channel $c$, the gradient of the attack loss can be expressed as

$$\nabla_{t_c}\mathcal{L}_{attack} = \mathbb{E}_{(x,\hat{y})\sim D_{bd}}[\frac{\partial\mathcal{L}_{CE}(f\theta(x'), \hat{y})}{\partial x'} \odot \frac{\partial x'}{\partial t_c}] \quad (7)$$

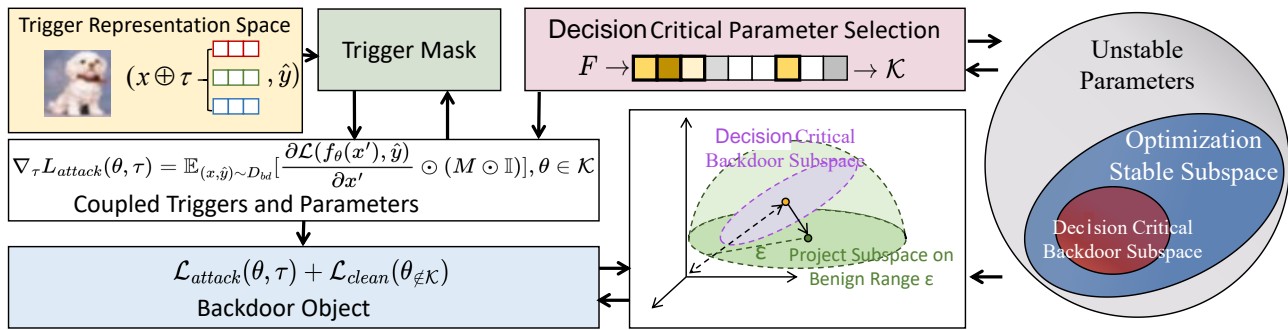

*Figure 1.* The overview of the FL backdoor attack framework.

where

$$\frac{\partial x'_{c,i,j}}{\partial t_{c,i,j}} = M_{c,i,j} \cdot \mathbb{I}(0 < x'_{c,i,j} < 1) \qquad (8)$$

and $\mathbb{I}(\cdot)$ denotes the indicator induced by the clipping operation. The trigger parameters are then updated by gradient descent with learning rate $\eta_\tau$.

This formulation encourages the trigger to act as a learnable input transformation that consistently shifts intermediate representations toward regions associated with the target class, while regularization through clean loss discourages reliance on unstable or overly salient artifacts.

### 4.2. Backdoor-Influential Stable Parameters

Federated learning implicitly suppresses parameters whose updates exhibit high variance across rounds and clients, while preserving directions with consistent gradient contributions. This motivates the need to identify a subset of parameters that are both influential for backdoor behavior and stable under the optimization dynamics of federated learning.

Let $\theta = \{w_i\}_{i=1}^p$ denote the model parameters. During federated training, each parameter $w_i$ undergoes a sequence of updates across communication rounds. We characterize the behavior of each parameter using two complementary properties: **optimization stability** and **backdoor responsiveness**.

We first quantify optimization stability by measuring the consistency of a parameter's updates across rounds. Let $w_i^{(t)}$ denote the value of parameter $w_i$ after aggregation at round $t$, and define the first-order update

$$\Delta w^{(t)} = w^{(t)} - w^{(t-1)} \qquad (9)$$

Over a window of $N$ consecutive rounds, we estimate the expected update

$$I_i = \mathbb{E}[\Delta w_i] \approx \frac{1}{N}\sum_{r=1}^{N} \Delta w_i^{(r)} \qquad (10)$$

which captures the average contribution of parameter $w_i$ to the optimization trajectory. Parameters with consistently non-negligible $I_i$ are less likely to be suppressed by aggregation and are therefore considered optimization-stable.

Next, we characterize backdoor responsiveness, which reflects how sensitively a parameter reacts to trigger-induced perturbations. Intuitively, parameters that exhibit high variability in response to poisoned inputs are more capable of amplifying backdoor signals. We estimate this responsiveness via the variance of parameter updates,

$$S_i = Var(\Delta w_i) \approx \frac{1}{N-1}\sum_{r=1}^{N}(\Delta w_i^{(r)} - I_i)^2 \qquad (11)$$

A larger $S_i$ indicates stronger responsiveness to input perturbations, including those induced by the trigger.

Highly stable parameters with negligible responsiveness contribute little to backdoor activation, while highly responsive but unstable parameters are likely to be attenuated by federated aggregation. To identify the optimal candidates for backdoor embedding, we define a **backdoor influence score** $F_i$ that characterizes the Signal-to-Noise Ratio (SNR) of each parameter by jointly accounting for both properties:

$$F_i = \left(\frac{|I_i|}{\sqrt{S_i} + \kappa}\right)^\alpha \qquad (12)$$

where $\kappa > 0$ is a small smoothing constant that prevents numerical instability when $\sqrt{S_i}$ is close to zero, and $\alpha > 0$ controls the sharpness of parameter selection. The score $F_i$ admits a signal-to-noise interpretation under federated aggregation: the numerator $|I_i|$ captures the coherent update signal of parameter $w_i$ across communication rounds, while the denominator $\sqrt{S_i}$ reflects stochastic and cross-client variation acting as aggregation-induced noise. Appendix A.2 formalizes this interpretation by showing that coordinates with higher effective SNR are more resistant to aggregation-induced attenuation. Consequently, a larger value of $F_i$ indicates that the corresponding parameter direction is more likely to be preserved and reinforced by

federated aggregation rather than canceled out by heterogeneous or unstable updates.

Based on the influence score $F_i$, we select the top $k\%$ parameters to form a decision-critical subset $\mathcal{K}$. This subset defines an optimization-stable subspace where backdoor signals can be embedded onto dominant benign update directions with reduced risk of aggregation-induced decay. To further ensure that the injected backdoor effect along these coordinates is not attenuated by subsequent federated aggregation, we apply a controlled scaling to the selected parameters, amplifying their effective contribution while keeping the update within prescribed bounds. The scaling factor follows the model replacement setting in (Bagdasaryan et al., 2020) and is used to compensate for the attenuation caused by federated averaging.

### 4.3. Coupled Subspace Optimization

Given the optimization-stable and backdoor-influential parameter subset $\mathcal{K}$ identified in Section 4.2, we formulate backdoor embedding as a constrained optimization problem that explicitly couples trigger-induced representations with stable parameter directions. The key idea is to restrict attack-driven updates to a low-dimensional subspace that is preserved by federated aggregation, while controlling the magnitude of parameter deviations to maintain stealth.

Let $\mathcal{K} \subset \{1 \cdots P\}$ index the selected parameter subset. During local training on malicious clients, the update rule is defined as

$$\Delta w_i = \begin{cases} -\eta \, \nabla_{w_i} \mathcal{L}_{attack}(\theta, \tau), & i \in \mathcal{K}, \\ -\eta \, \nabla_{w_i} \mathcal{L}_{clean}(\theta), & i \notin \mathcal{K}. \end{cases} \quad (13)$$

This selective update enforces an explicit coupling between trigger optimization and parameter manipulation, ensuring that trigger-induced gradients propagate only through optimization-stable directions.

From an optimization perspective, Eq. (13) corresponds to projecting the attack-driven gradient onto the subspace spanned by $\mathcal{K}$. Let $\mathcal{P}_{\mathcal{K}}$ denote the associated projection operator. The malicious update can be compactly expressed as

$$\Delta\theta = -\eta \, \mathcal{P}_{\mathcal{K}}\big(\nabla_\theta \mathcal{L}_{attack}(\theta, \tau)\big) - \eta \, \mathcal{P}_{\bar{\mathcal{K}}}\big(\nabla_\theta \mathcal{L}_{clean}(\theta)\big). \quad (14)$$

leading to the projected update

$$w_i^{new} = Proj_{||\cdot||_\infty}(w_i^{old} + \Delta w_i), i \in \mathcal{K} \quad (15)$$

Combining the subspace restriction and the norm constraint, the backdoor embedding process can be viewed as solving

$$min \mathcal{L}_{attack}(\theta, \tau) \quad s.t. \quad supp(\Delta\theta) \subseteq \mathcal{K}, ||\Delta w_i||_\infty \leq \epsilon \quad (16)$$

---

**Algorithm 1** Training Procedure of CTO-VPA

**Input:** $\theta^G, D_{\text{cl}}, D_{\text{bd}}, N, k, \eta, \eta_\tau, \lambda, \epsilon, \kappa, \alpha$
**Output:** $\Delta\theta, \tau, K$

1: $\theta \leftarrow \theta^G, \quad \tau \leftarrow \{t_r, t_g, t_b\}$
2: **for** $i = 1, \ldots, p$ **do**
3: $\quad \Delta w_i^{(t)} \leftarrow w_i^{(t)} - w_i^{(t-1)}$ $\qquad\qquad \triangleright$ Eq. (9)
4: $\quad I_i \leftarrow \frac{1}{N}\sum_{r=1}^{N} \Delta w_i^{(r)}$ $\qquad\qquad \triangleright$ Eq. (10)
5: $\quad S_i \leftarrow \frac{1}{N-1}\sum_{r=1}^{N}\left(\Delta w_i^{(r)} - I_i\right)^2$ $\quad \triangleright$ Eq. (11)
6: $\quad F_i \leftarrow \left(\frac{|I_i|}{\sqrt{S_i}+\kappa}\right)^\alpha$ $\qquad\qquad \triangleright$ Eq. (12)
7: **end for**
8: $K \leftarrow \text{TopK}\left(\{F_i\}_{i=1}^p, k\%\right)$
9: **for** $s = 1, \ldots, S$ **do**
10: $\quad x'_{c,i,j} \leftarrow \text{clip}\left(x_{c,i,j} + M_{i,j} \odot t_c\right), \quad c \in \{r, g, b\}$
$\quad \triangleright$ Eq. (4)
11: $\quad L_{\text{attack}}(\theta, \tau) \leftarrow \mathbb{E}_{(x,\hat{y})\sim D_{\text{bd}}}\left[L_{\text{CE}}\left(f_\theta(x'), \hat{y}\right)\right]$
12: $\quad L_{\text{clean}}(\theta) \leftarrow \mathbb{E}_{(x,y)\sim D_{\text{cl}}}\left[L_{\text{CE}}\left(f_\theta(x), y\right)\right]$
13: $\quad \tau \leftarrow \tau - \eta_\tau \nabla_\tau L_{\text{attack}}(\theta, \tau)$
14: $\quad$ **for** $i = 1, \ldots, p$ **do**
15: $\qquad \Delta w_i \leftarrow \begin{cases} -\eta \nabla_{w_i} L_{\text{attack}}(\theta, \tau), & i \in K, \\ -\eta \nabla_{w_i} L_{\text{clean}}(\theta), & i \notin K. \end{cases} \triangleright$
$\quad$ Eq. (13)
16: $\quad$ **end for**
17: $\quad \Delta\theta \leftarrow -\eta P_K \nabla_\theta L_{\text{attack}}(\theta, \tau) - \eta P_{\bar{K}} \nabla_\theta L_{\text{clean}}(\theta)$
$\quad \triangleright$ Eq. (14)
18: $\quad \theta \leftarrow \theta + \Delta\theta$
19: $\quad$ **for** $i \in K$ **do**
20: $\qquad w_i^{\text{new}} \leftarrow \text{Proj}_{\|\cdot\|_\infty}\left(w_i^{\text{old}} + \Delta w_i\right) \quad \triangleright$ Eq. (15)
21: $\quad$ **end for**
22: **end for**
23: $\Delta\theta \leftarrow \theta - \theta^G$
24: **return** $\Delta\theta, \tau, K$

---

This formulation highlights that persistence and stealth are jointly achieved by embedding backdoor behavior into optimization-stable directions with bounded perturbations. Because federated aggregation naturally preserves stable directions, even constrained updates can accumulate coherently across rounds, enabling long-lasting backdoor effects. Appendix A.1 shows that the coupled trigger-parameter update can be interpreted as a projected alternating descent procedure, while Appendix A.3 bounds the aggregation gap induced by benign updates within the selected subspace.

The overall training procedure of CTO-VPA is summarized in Algorithm 1. The algorithm first estimates coordinate-wise update statistics over an $N$-round observation window, then selects the top-$k\%$ stable and backdoor-influential coordinates, and finally performs coupled trigger and parameter optimization under a bounded subspace projection.

# 5. Experimental Evaluation

## 5.1. Experiment Setup

The proposed backdoor attack was conducted using PyTorch and Python 3 on a multi-GPU system equipped with 4× NVIDIA GeForce RTX 5090 GPUs (32GB VRAM each), accompanied by an AMD EPYC 9654 48-Core CPU and 125GB instance memory.

We conduct the experiments on three datasets: CIFAR-10 (Krizhevsky & Hinton, 2009), GTSRB (Stallkamp et al., 2011), and Tiny-ImageNet (Le & Yang, 2015). For CIFAR-10 and GTSRB, we use ResNet-18 (He et al., 2016) as the backbone model. For Tiny-ImageNet, we employ Astroformer (Dagli, 2023) as the backbone model to further validate the effectiveness of our attack on more complex datasets and stronger architectures.

**Backdoor Attacks.** We compare the `CTO-VPA` with five SOTA backdoor attack methods: 1. constrained backdoor attack (CBA) (Sun et al., 2019), 2. distributed backdoor attack (DBA) (Xie et al., 2019), 3. durable backdoor attack (Neurotoxin) (Zhang et al., 2022), 4. fully-combination backdoor attack (FCBA) (Liu et al., 2024), 5. personal backdoor attack (PFL) (Fan et al., 2025). We assume that malicious clients constitute 10% of all participants in each communication round. For all kinds of backdoor, we set the poisoned data rate to 25%. The top-k ratio of our subspace is 30%, the sharpness parameter is $\alpha = 1.5$, the window length is $N = 30$.

**Evaluation Metric.** We evaluate the performance of the proposed backdoor attack on the following three metrics: Main Task Accuracy (MA), Backdoor Accuracy (BA), and Lifespan.

**Attack Schedules.** We consider three attack schedules to evaluate both the effectiveness and persistence of the proposed method. In the **single-shot** setting, the malicious client injects the backdoor update only once at a single global round and then remains benign in all subsequent rounds. In **multi-shot 1**, the attacker periodically injects the backdoor update once every 10 global rounds, simulating sporadic but repeated poisoning. In **multi-shot 2**, the attacker injects the backdoor update at every global round, corresponding to a continuous attack scenario.

## 5.2. Attack Efficacy

We compare `CTO-VPA` with representative FL backdoor attacks under single-shot and multi-shot settings. The **single-shot** setting is most indicative of backdoor persistence, as the attack must survive subsequent benign aggregation without reinforcement. As shown in Table 1, `CTO-VPA` achieves substantially higher BA across all datasets (71.43% on CIFAR-10, 51.50% on GTSRB, and 50.25% on Tiny-

ImageNet) while maintaining competitive MA. In contrast, data-level attacks such as CBA, DBA, and FCBA exhibit limited or near-zero BA after aggregation, highlighting their vulnerability to dilution by benign updates. These results are consistent with our design by coupling trigger-induced representations with optimization-stable, backdoor-influential parameters, `CTO-VPA` embeds backdoor behavior into directions naturally preserved by federated optimization.

Under periodic and continuous poisoning (**multi-shots**), most methods benefit from repeated reinforcement. Nevertheless, `CTO-VPA` consistently reaches near-saturated BA with negligible MA degradation, whereas several baselines show either lower BA or unstable performance across datasets. This suggests that `CTO-VPA` improves update efficiency: each poisoning round contributes coherently within the same stable subspace, reducing the need for aggressive or widespread parameter manipulation.

Notably, some model-level attacks can achieve high BA under continuous poisoning but at the cost of severe MA degradation, indicating poor stealth. `CTO-VPA` avoids this trade-off by constraining attack-driven updates to a bounded, optimization-stable subspace, enabling reliable backdoor activation while preserving benign performance. Overall, the results demonstrate that explicit trigger–parameter coupling is critical for achieving both persistent and stealthy backdoor attacks in federated learning.

## 5.3. Backdoor Longevity Under Single-Shot Attacks

Figure 2 evaluates backdoor longevity under the single-shot setting, where persistence is measured by the number of rounds before BA drops below 50%. `CTO-VPA` consistently exhibits the slowest BA decay across datasets. On CIFAR-10, its BA remains above the threshold for over 250 rounds, whereas all baseline methods fall below 50% within tens of rounds. This significant difference suggests that `CTO-VPA` embeds backdoor behavior into parameter directions that are preserved by federated aggregation, rather than relying on transient perturbations that are quickly neutralized by benign updates. Importantly, the extended lifespan does not come at the cost of degraded main-task performance. While some baselines either lose BA rapidly or incur noticeable MA drops, `CTO-VPA` maintains stable MA alongside sustained BA.

## 5.4. Backdoor Attack against Defenses

Table 2 compares robustness under magnitude-based (Norm-Clipping), robust aggregation (Trimmed-Mean), stochastic perturbation (DP), and anomaly-based (FLAME) defenses. Across all datasets, `CTO-VPA` consistently maintains near-perfect BA under NormClipping, Trimmed-Mean, and DP (BA $\approx 100$ on CIFAR-10 and GTSRB, and above 98 on Tiny-ImageNet), while preserving MA close to be-

*Table 1.* Performance Comparison of Backdoor Attack Methods across Datasets.

| Method | CIFAR | | | | | | GTSRB | | | | | | Tiny-ImageNet | | | | | |
|---|---|---|---|---|---|---|---|---|---|---|---|---|---|---|---|---|---|---|
| | Single-shot | | Multi-shot1 | | Multi-shot2 | | Single-shot | | Multi-shot1 | | Multi-shot2 | | Single-shot | | Multi-shot1 | | Multi-shot2 | |
| | MA↑ | BA↑ | MA↑ | BA↑ | MA↑ | BA↑ | MA↑ | BA↑ | MA↑ | BA↑ | MA↑ | BA↑ | MA↑ | BA↑ | MA↑ | BA↑ | MA↑ | BA↑ |
| CBA | 90.64 | 2.72 | 88.96 | 94.16 | 79.83 | 97.64 | 96.47 | 0.23 | 90.07 | 91.51 | 98.33 | 85.10 | 50.20 | 0.25 | 50.47 | 68.9 | **53.56** | 93.06 |
| DBA | 90.58 | 8.30 | 89.42 | 94.58 | 85.97 | 97.80 | 96.40 | 18.24 | 96.00 | 85.79 | 98.21 | 86.28 | 49.48 | 0.26 | 51.42 | 51.51 | 53.16 | 88.91 |
| Neurotoxin | 88.61 | 1.17 | 86.58 | 58.59 | 42.84 | **100** | 96.39 | 0.00 | 95.69 | 63.28 | 97.74 | 87.70 | **52.89** | 0.80 | 10.61 | 3.80 | 16.72 | 59.80 |
| FCBA | 90.61 | 10.66 | 89.47 | 96.69 | 86.61 | 98.93 | 96.51 | 18.70 | 96.16 | 89.70 | **98.46** | 85.94 | 50.79 | 0.27 | **52.21** | 47.26 | 53.10 | 91.58 |
| PFL | **90.71** | 33.72 | 89.35 | 97.76 | **90.55** | 99.96 | 96.72 | 3.86 | 94.48 | 95.02 | 98.51 | 90.96 | 49.70 | 21.09 | 50.89 | 98.13 | 48.99 | 99.68 |
| CTO-VPA | 90.60 | **71.43** | **89.48** | **100** | 90.03 | **100** | **97.30** | **51.50** | **96.24** | **100** | 98.28 | **100** | 50.57 | **50.25** | 51.13 | **99.90** | 53.14 | **99.88** |

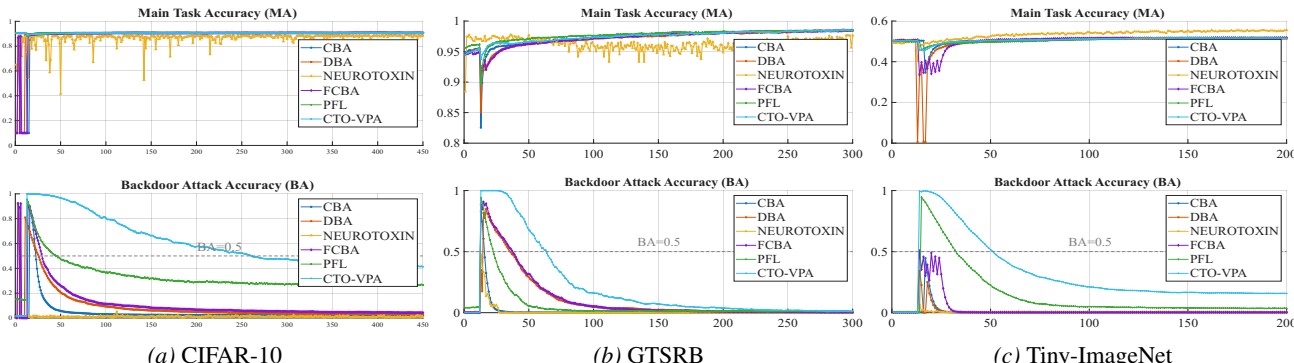

*(a)* CIFAR-10      *(b)* GTSRB      *(c)* Tiny-ImageNet

*Figure 2.* The comparison of the main task accuracy and backdoor longevity under single-shot attacks.

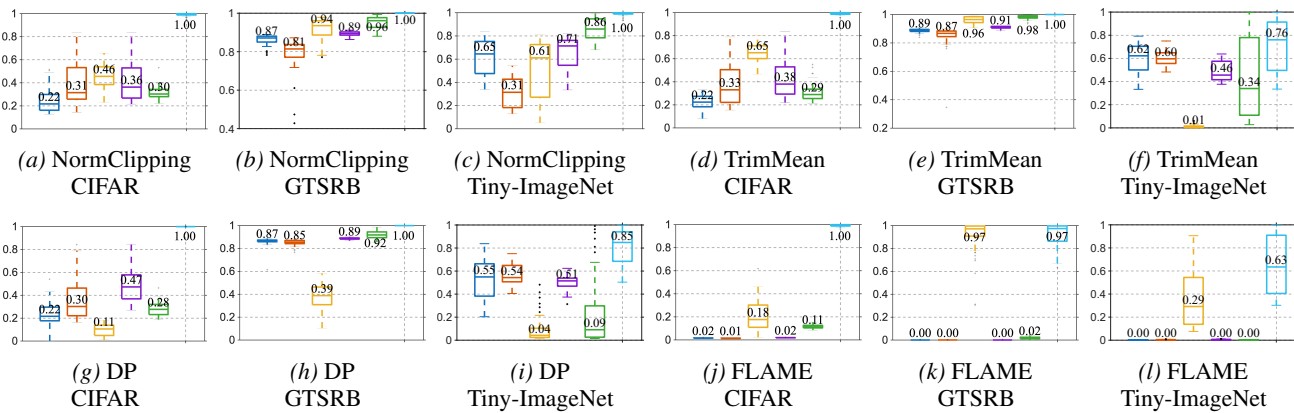

*(a)* NormClipping CIFAR   *(b)* NormClipping GTSRB   *(c)* NormClipping Tiny-ImageNet   *(d)* TrimMean CIFAR   *(e)* TrimMean GTSRB   *(f)* TrimMean Tiny-ImageNet

*(g)* DP CIFAR   *(h)* DP GTSRB   *(i)* DP Tiny-ImageNet   *(j)* FLAME CIFAR   *(k)* FLAME GTSRB   *(l)* FLAME Tiny-ImageNet

*Figure 3.* Comparison of backdoor persistence under different defense strategies across datasets. In each figure from left to right the methods are CBA, DBA, Neurotoxin, FCBA, PFL and CTO-VPA.

nign training. In contrast, data-trigger baselines such as CBA/DBA/FCBA show noticeably lower and less stable BA under these defenses, indicating sensitivity to update attenuation and injected noise rather than inherent robustness.

FLAME further reveals a sharp separation among methods. While it nearly eliminates backdoors for CBA, DBA, and FCBA across datasets (BA ≈ 1 ∼ 2% on CIFAR-10 and near zero on GTSRB), CTO-VPA remains highly effective with BA close to 100% and no observable MA degradation.

*Table 2.* Performance Comparison of Backdoor Attack Methods under Different Defenses.

| Method | CIFAR | | | | | | | | GTSRB | | | | | | | | Tiny-ImageNet | | | | | | | |
|---|---|---|---|---|---|---|---|---|---|---|---|---|---|---|---|---|---|---|---|---|---|---|---|---|
| | NormClipping | | Trimmed-Mean | | DP | | FLAME | | NormClipping | | Trimmed-Mean | | DP | | FLAME | | NormClipping | | Trimmed-Mean | | DP | | FLAME | |
| | MA↑ | BA↑ | MA↑ | BA↑ | MA↑ | BA↑ | MA↑ | BA↑ | MA↑ | BA↑ | MA↑ | BA↑ | MA↑ | BA↑ | MA↑ | BA↑ | MA↑ | BA↑ | MA↑ | BA↑ | MA↑ | BA↑ | MA↑ | BA↑ |
| CBA | 89.99 | 85.38 | 90.05 | 78.70 | 90.07 | 87.12 | 90.49 | 1.52 | 98.29 | 86.90 | 98.23 | 85.21 | 98.75 | 86.46 | 98.39 | 0.07 | 49.07 | 95.40 | 49.44 | 98.50 | 49.00 | 96.13 | 46.96 | 0.21 |
| DBA | 90.14 | 86.58 | 90.15 | 88.14 | 89.86 | 87.52 | 90.35 | 1.64 | 98.48 | 79.94 | 98.30 | 86.14 | 98.42 | 82.91 | 98.29 | 0.07 | 49.46 | 91.07 | 49.83 | 92.75 | 48.81 | 91.38 | 47.49 | 0.28 |
| Neurotoxin | 70.94 | 49.80 | 72.02 | 66.80 | 76.98 | 11.91 | 38.88 | 32.93 | 98.01 | 78.13 | 97.76 | 88.28 | 84.16 | 13.67 | 93.73 | 100 | 42.25 | 78.20 | 13.01 | 62.80 | 48.62 | 42.20 | 33.58 | 84.80 |
| FCBA | 90.35 | 90.62 | 90.07 | 90.98 | 90.70 | 88.11 | 90.37 | 1.84 | 98.70 | 87.22 | 98.84 | 90.54 | 98.78 | 88.40 | 98.41 | 0.09 | 51.16 | 92.54 | 50.74 | 90.55 | 51.01 | 91.14 | 47.75 | 0.48 |
| PFL | 90.20 | 87.37 | 90.16 | 84.81 | 90.12 | 81.18 | 90.22 | 12.10 | 98.61 | 88.53 | 98.43 | 94.50 | 98.66 | 90.56 | 98.34 | 2.85 | 50.30 | 99.83 | 49.02 | 99.73 | 48.33 | 98.09 | 46.96 | 0.21 |
| CTO-VPA | 90.31 | **100** | 90.19 | **100** | 90.51 | **100** | 90.22 | **100** | 98.47 | **100** | 98.50 | **100** | 98.66 | **100** | 98.24 | 98.69 | 49.00 | **100** | 50.18 | 99.91 | 50.77 | 98.86 | 47.06 | 99.86 |

Neurotoxin partially survives FLAME on specific datasets but degrades substantially elsewhere, reflecting unstable behavior across tasks. These results suggest that robustness against strong anomaly-based defenses is primarily determined by where and how backdoor signals are embedded. `CTO-VPA`'s constrained optimization within a stable, decision-critical subspace yields updates that are both distributionally benign and functionally effective, enabling consistent robustness across diverse defenses.

### 5.5. Persistence Under Defense

Figure 3 isolates post-attack robustness by tracking BA over 50 consecutive rounds after poisoning has stopped, while defenses are applied every round. This figure reveals the stability of the implanted backdoor: `CTO-VPA` maintains an almost flat BA trajectory under NormClipping, Trimmed-Mean, and DP, indicating that the backdoor is not sustained by continued reinforcement but is embedded in aggregation-resistant components that remain effective under repeated defensive perturbations.

In contrast, existing attacks show pronounced instability over the 50 rounds, as evidenced by large gaps between their maximum, median, and minimum BA. This variability indicates reliance on brittle mechanisms, such as transient gradient amplification or fragile parameter perturbations, which can be weakened or intermittently disrupted by robust aggregation and noise injection. Neurotoxin exhibits comparatively stronger persistence, especially under FLAME, consistent with selective model-level poisoning, but its higher variability across defenses and datasets highlights the limitations of lacking explicit coupling and constraint. As a result, the backdoor remains more sensitive to stochastic perturbations and less consistent over long horizons.

### 5.6. Robustness under Data Heterogeneity

We further evaluate CTO-VPA under Dirichlet label skew with $\alpha \in \{0.1, 0.5, 0.9\}$, where a smaller $\alpha$ indicates stronger client heterogeneity. As shown in Table 3, CTO-VPA remains effective under different degrees of label skew. Under the more extreme heterogeneity setting ($\alpha = 0.1$), CTO-VPA exhibits a clear persistence advantage over the baselines, maintaining a slower BA decay after the attack stops. Under the moderate non-IID setting ($\alpha = 0.5$), CTO-VPA performs comparably to PFL. This is expected because PFL is designed as a personalized backdoor attack and is therefore naturally better aligned with non-IID scenarios where client distributions are more distinguishable. Even in this case, CTO-VPA remains competitive and maintains strong post-attack persistence. These results suggest that the proposed attack is not restricted to IID or simplified settings, but remains effective under more realistic heterogeneous client distributions.

*Table 3.* Backdoor performance under data heterogeneity.

| Method | $\alpha = 0.1$ | $\alpha = 0.5$ | $\alpha = 0.9$ | **Lifespan** $\lambda$ |
|---|---|---|---|---|
| CBA | 59.43 | 83.28 | 54.92 | 2/11/2 |
| DBA | 87.63 | 95.98 | 87.40 | 13/26/4 |
| Neurotoxin | 64.65 | 100.00 | 100.00 | 5/8/1 |
| FCBA | 99.98 | 94.99 | 92.81 | 22/80/16 |
| PFL | 76.38 | 100.00 | 94.60 | 8/200/20 |
| **CTO-VPA** | **92.31** | 98.87 | 99.27 | **71/173/65** |

### 5.7. Ablation Study

**Ablation on Trigger–Parameter Coupling:** Figure 4a provides a controlled ablation that isolates the **model-poisoning strategy** by fixing the trigger to DBA and running the attack only from rounds 14 to 113 before completely stopping. A key observation is that both methods achieve comparable BA during the active poisoning phase, which suggests that

short-term attack strength is not the main differentiator here. The divergence emerges immediately after the attack ceases, where Neurotoxin's BA drops rapidly. This behavior indicates that Neurotoxin's backdoor largely relies on continued, high-amplitude parameter perturbations that are quickly "re-optimized away" by subsequent benign aggregation once reinforcement disappears.

In contrast, CTO-VPA maintains a markedly slower decay in the post-attack period despite sharing the same trigger and attack window. This supports the interpretation that persistence is governed by **where the backdoor is encoded** rather than how aggressively parameters are perturbed. By concentrating attack gradients on a stable, backdoor-influential subset and constraining update magnitude, CTO-VPA embeds the backdoor into parameter directions that remain effective under continued benign training, leading to sustained BA even without further poisoning. Fig. 4b substantiates that selective, constrained subspace embedding is critical for persistence beyond the attack phase, whereas unconstrained amplification yields brittle, short-lived backdoors.

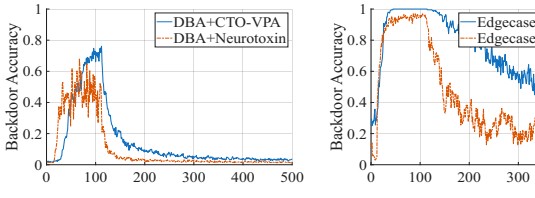

*(a)* Different model poisoning strategies

*(b)* Different trigger designs

*Figure 4.* Backdoor success rate under edge-case poisoning with different model poisoning strategies and trigger designs.

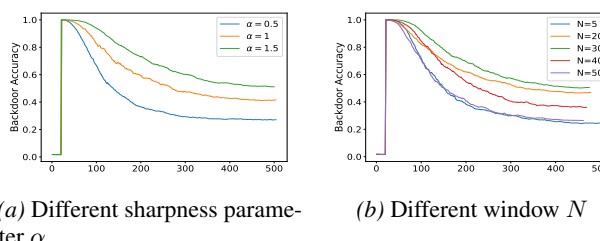

*(a)* Different sharpness parameter $\alpha$

*(b)* Different window $N$

*Figure 5.* Ablation on parameter $\alpha$ and $N$.

**Ablation on the Sharpness Parameter** $\alpha$**:** Figure 5a reports the backdoor accuracy (BA) under different $\alpha$ in Eq. (12). A larger $\alpha$ consistently slows down the post-attack decay and yields a higher long-horizon BA. Specifically, $\alpha = 1.5$ maintains the strongest persistence (BA $\approx 0.51$ at round 500), while $\alpha = 1$ and $\alpha = 0.5$ decay to BA $\approx 0.41$ and $\approx 0.27$, respectively. This trend indicates that increasing $\alpha$ sharpens the selection toward coordinates that are simultaneously more stable and more backdoor-responsive, thereby reducing aggregation-induced attenuation. Importantly, the attack remains effective across a wide range of $\alpha$, suggest-

ing that $\alpha$ mainly controls the separation strength rather than being a sensitive tuning parameter. From an optimization perspective, a larger $\alpha$ suppresses marginal coordinates with moderate signal-to-noise ratios, concentrating the attack update on a smaller set of aggregation-favored directions. As a result, the accumulated backdoor signal experiences less cancellation under subsequent benign updates, leading to improved persistence.

**Ablation on the Window** $N$**:** We conduct a single-shot backdoor attack to examine the stability window length $N$ used in estimating the coordinate-wise influence statistics. In this setting, the adversary injects the malicious update only once and remains benign thereafter, making the subsequent decay of backdoor accuracy solely determined by how well the implanted signal survives repeated benign aggregation. As shown in Fig. 5b, a small window ($N = 5$) leads to rapid post-attack degradation, indicating that short-horizon update estimates are dominated by stochastic fluctuations and fail to reliably capture aggregation-preserved directions. Increasing $N$ substantially improves persistence, with intermediate values ($N = 20 \sim 30$) achieving long-term backdoor accuracy. This regime provides a favorable trade-off where the coherent update signal $I_i$ is sufficiently smoothed while maintaining sensitivity to trigger-responsive variability $S_i$. In contrast, overly large windows ($N = 50$) slightly weaken persistence, suggesting that excessive averaging may dilute decision-critical backdoor-responsive coordinates and bias selection toward purely stable but weakly influential parameters.

## 6. Conclusions

In this work, we study backdoor attacks in federated learning from an optimization perspective and showed that long-term persistence depends on the alignment between trigger-induced representations and parameter directions that remain stable under aggregation. Building on this insight, we proposed CTO-VPA, which couples trigger optimization with constrained updates in a decision-critical and optimization-stable subspace. This formulation enables backdoor behaviors to survive FL aggregation and defenses without degrading benign performance. Extensive experiments across multiple datasets, attack schedules, and defense mechanisms demonstrate that CTO-VPA achieves substantially improved persistence and robustness compared to existing attacks. Our analysis highlights that backdoor persistence is not merely a function of attack intensity, but depends on where and how malicious behavior is embedded in the model. This contributes to a deeper understanding of optimization dynamics in federated learning and informs the design of more principled defenses against persistent backdoor threats.

## Acknowledgements

This research is supported in part by the National Key Research and Development Program of China (2023YFB3107505), the Shaanxi Provincial Outstanding Youth Science Fund (2023-JC-JQ-52), the National Natural Science Foundation of China (62302371), the State-funded Postdoctoral Researcher Program (GZC20232035), the China Scholarship Council (202506960039), and the Ministry of Education, Singapore, under its Academic Research Fund Tier 1 (RG101/24).

## Impact Statement

This work analyzes persistent backdoor vulnerabilities in federated learning by showing how malicious behaviors can be embedded into optimization-stable parameter subspaces favored by aggregation. Such understanding is important for strengthening the security of federated systems deployed in privacy-sensitive domains. We acknowledge the dual-use nature of backdoor research, as similar techniques could be misused by adversaries. Our goal is to expose these failure modes to motivate more robust defenses.

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

# A. Theoretical Analysis

## A.1. Convergence Analysis of the Coupled Projected Optimization

We analyze the optimization dynamics of CTO-VPA as a projected alternating descent procedure. The algorithm alternates between updating the trigger $\tau$ and updating the model parameters $\theta$ under different objectives and constraints, as specified in Eq. (13)–(16) of the main paper.

At iteration $t$, the updates are given by

$$\tau^{t+1} = \tau^t - \eta_\tau \nabla_\tau \mathcal{L}_{\text{attack}}(\theta^t, \tau^t), \tag{17}$$

and

$$\theta^{t+1} = \Pi_{\mathcal{C}_\epsilon}\big(\theta^t - \eta\, g(\theta^t, \tau^{t+1})\big), \tag{18}$$

where

$$g(\theta, \tau) = P_{\mathcal{K}} \nabla_\theta \mathcal{L}_{\text{attack}}(\theta, \tau) + P_{\overline{\mathcal{K}}} \nabla_\theta \mathcal{L}_{\text{clean}}(\theta), \tag{19}$$

and the constraint set is

$$\mathcal{C}_\epsilon = \Big\{ \theta \mid \|P_{\mathcal{K}}(\theta - \theta^0)\|_\infty \le \epsilon \Big\}. \tag{20}$$

We make the following assumptions.

**Assumption A.1.** The loss functions $\mathcal{L}_{\text{attack}}(\theta, \tau)$ and $\mathcal{L}_{\text{clean}}(\theta)$ are continuously differentiable and $L_\theta$-smooth with respect to $\theta$, and $\mathcal{L}_{\text{attack}}(\theta, \tau)$ is $L_\tau$-smooth with respect to $\tau$.

**Lemma A.2** (Descent of the Parameter Update). *For a fixed trigger $\tau$, let*

$$\theta^+ = \Pi_{\mathcal{C}_\epsilon}(\theta - \eta g(\theta, \tau)),$$

*with $\eta \le 1/L_\theta$. Then the gradient mapping*

$$G_\eta(\theta, \tau) \triangleq \frac{1}{\eta}\big(\theta - \theta^+\big)$$

*satisfies*

$$\langle g(\theta, \tau), G_\eta(\theta, \tau) \rangle \ge \|G_\eta(\theta, \tau)\|^2. \tag{21}$$

*Proof.* The result follows directly from the optimality condition of the Euclidean projection onto the closed convex set $\mathcal{C}_\epsilon$ and standard properties of projected gradient methods. $\square$

**Lemma A.3** (Sufficient Decrease of a Lyapunov Function). *Define the composite function*

$$\Psi(\theta, \tau) = \mathcal{L}_{\text{attack}}(\theta, \tau) + \mathcal{L}_{\text{clean}}(\theta).$$

*Under Assumption 1 and $\eta \le 1/L_\theta$, the update in Eq. (18) yields*

$$\Psi(\theta^{t+1}, \tau^{t+1}) \le \Psi(\theta^t, \tau^{t+1}) - \frac{\eta}{2}\|G_\eta(\theta^t, \tau^{t+1})\|^2. \tag{22}$$

*Proof.* By $L_\theta$-smoothness of $\Psi$ with respect to $\theta$ and applying Lemma A.2, the standard descent inequality for projected gradient steps applies. $\square$

**Theorem A.4** (Convergence to a Stationary Point). *Let $\{\theta^t, \tau^t\}$ be the sequence generated by Eq. (17)–(18). If $\eta \le 1/L_\theta$ and $\eta_\tau \le 1/L_\tau$, then*

$$\frac{1}{T} \sum_{t=0}^{T-1} \mathbb{E}\big[\|G_\eta(\theta^t, \tau^t)\|^2\big] \le \mathcal{O}\Big(\frac{1}{T}\Big). \tag{23}$$

*Consequently, the projected alternating descent dynamics of CTO-VPA converges to a first-order stationary point.*

*Proof.* By Lemma A.3, the sequence $\{\Psi(\theta^t, \tau^t)\}$ is non-increasing and bounded below. Summing the sufficient decrease inequality over $t = 0, \ldots, T-1$ yields a telescoping series, from which the result follows. $\square$

## A.2. Backdoor Persistence via Coordinate-wise Signal-to-Noise Ratio

We analyze the persistence of backdoor signals under federated aggregation from a coordinate-wise signal-to-noise perspective.

**Assumption A.5** (Benign Update Statistics). For each parameter coordinate $i$, the benign aggregated update $\Delta_{\text{benign},i}$ is a random variable with finite mean $I_i$ and finite variance $S_i$, i.e.,

$$\mathbb{E}[\Delta_{\text{benign},i}] = I_i, \qquad \text{Var}(\Delta_{\text{benign},i}) = S_i. \tag{24}$$

**Definition A.6** (Aggregated Coordinate Update). Let $\Delta_i$ denote the aggregated update along coordinate $i$ at a given federated round, defined as

$$\Delta_i = \rho\,\Delta_{\text{adv},i} + (1 - \rho)\,\Delta_{\text{benign},i}, \tag{25}$$

where $\rho \in (0, 1)$ is the effective aggregation weight of the adversarial clients.

**Definition A.7** (Effective Signal-to-Noise Ratio). The effective signal-to-noise ratio along coordinate $i$ is defined as

$$\text{SNR}_i \triangleq \frac{|\mathbb{E}[\Delta_i]|}{\sqrt{\text{Var}(\Delta_i)}}. \tag{26}$$

**Lemma A.8** (Monotonicity of SNR). *For fixed $\rho$, $\Delta_{\text{adv},i}$, and $I_i$, the quantity $\text{SNR}_i$ is a strictly decreasing function of $S_i$.*

*Proof.* By Assumption 5, $\text{Var}(\Delta_i) = (1 - \rho)^2 S_i$. For fixed numerator $|\mathbb{E}[\Delta_i]|$, the denominator $\sqrt{\text{Var}(\Delta_i)}$ is strictly increasing in $S_i$ for $S_i > 0$, which implies the claim. $\square$

**Lemma A.9** (Relative Preservation under Aggregation). *Consider two coordinates $i$ and $j$ satisfying*

$$\Delta_{\text{adv},i} = \Delta_{\text{adv},j}, \qquad I_i = I_j.$$

*If $S_i < S_j$, then $\text{SNR}_i > \text{SNR}_j$.*

*Proof.* The result follows immediately from Lemma A.8 by comparing the denominators of $\text{SNR}_i$ and $\text{SNR}_j$. $\square$

**Theorem A.10** (SNR-based Parameter Selection and Backdoor Persistence). *Let the influence score be defined as*

$$F_i = \left(\frac{|I_i|}{\sqrt{S_i} + \kappa}\right)^{\alpha},$$

*and let $\mathcal{K}$ denote the set of coordinates selected by the top-$k\%$ values of $\{F_i\}$. Then, for any $i \in \mathcal{K}$ and $j \notin \mathcal{K}$ with comparable adversarial injection magnitudes, the aggregated backdoor signal along coordinate $i$ is more resistant to attenuation by federated aggregation than that along coordinate $j$.*

*Proof.* By construction, selection by $F_i$ prioritizes coordinates with larger ratios $|I_i|/(\sqrt{S_i} + \kappa)$. Since $\kappa$ is constant, this ordering is consistent with that induced by $|I_i|/\sqrt{S_i}$. By Lemma A.9, coordinates with smaller $S_i$ preserve injected signals more effectively under aggregation. Therefore, restricting adversarial updates to $\mathcal{K}$ yields higher persistence relative to unselected coordinates. $\square$

This coordinate-wise analysis provides a tractable proxy for the spectral properties of the benign update distribution, ensuring that backdoor signals are embedded in the high-SNR manifold of the model's loss landscape.

## A.3. Aggregation Gap Bound under Variance-Constrained Subspace

We analyze the discrepancy between the attacker's intended update and the actual aggregated update under federated aggregation, referred to as the aggregation gap.

**Definition A.11** (Aggregation Gap). Let $\Delta_{\text{adv}} \in \mathbb{R}^d$ denote the adversarial update constructed by the attacker, and let

$$\Delta = \rho\,\Delta_{\text{adv}} + (1 - \rho)\,\Delta_{\text{benign}}$$

be the aggregated update at the server. For a loss function $\mathcal{L}_{\text{attack}}(\theta)$, the aggregation gap is defined as

$$G \triangleq \mathbb{E}[|\mathcal{L}_{\text{attack}}(\theta + \Delta_{\text{adv}}) - \mathcal{L}_{\text{attack}}(\theta + \Delta)|]. \tag{27}$$

**Assumption A.12** (Smoothness). The loss function $\mathcal{L}_{\text{attack}}(\theta)$ is $L$-smooth with respect to $\theta$.

**Assumption A.13** (Benign Update Statistics). For each coordinate $i$, the benign aggregated update $\Delta_{\text{benign},i}$ has mean $I_i$ and variance $S_i$, and the coordinates are uncorrelated.

**Lemma A.14** (Quadratic Upper Bound on Loss Deviation). *Under Assumption A.12, for any vectors $u, v \in \mathbb{R}^d$,*

$$|\mathcal{L}_{\text{attack}}(\theta + u) - \mathcal{L}_{\text{attack}}(\theta + v)| \leq \langle \nabla \mathcal{L}_{\text{attack}}(\theta + v),\, u - v \rangle + \frac{L}{2}\|u - v\|_2^2. \tag{28}$$

*Proof.* The result follows directly from the second-order Taylor expansion with $L$-smoothness. $\square$

**Lemma A.15** (Expected Squared Deviation). *Let $Z \triangleq \Delta_{\text{adv}} - \Delta$. Under Assumption A.13,*

$$\mathbb{E}\big[\|Z\|_2^2\big] = (1 - \rho)^2 \sum_{i=1}^{d} S_i. \tag{29}$$

*Proof.* By definition,
$$Z = (1 - \rho)(\Delta_{\text{adv}} - \Delta_{\text{benign}}).$$

Since $\mathbb{E}[\Delta_{\text{benign},i}] = I_i$ and $\Delta_{\text{adv}}$ is deterministic,

$$\mathbb{E}\big[(\Delta_{\text{adv},i} - \Delta_{\text{benign},i})^2\big] = \text{Var}(\Delta_{\text{benign},i}) = S_i.$$

Summing over coordinates and using the uncorrelated assumption yields the claim. $\square$

**Theorem A.16** (Tight Aggregation Gap Bound). *Suppose Assumptions A.12–A.13 hold and $\Delta_{\text{adv}}$ is supported on a coordinate subset $\mathcal{K}$. Then the aggregation gap satisfies*

$$G \;\leq\; \frac{L}{2}(1 - \rho)^2 \sum_{i \in \mathcal{K}} S_i + (1 - \rho)\sqrt{\sum_{i \in \mathcal{K}} S_i}\; \mathbb{E}[\|\nabla_{\mathcal{K}}\mathcal{L}_{\text{attack}}(\theta + \Delta)\|_2], \tag{30}$$

*where $\nabla_{\mathcal{K}}$ denotes the gradient restricted to the subspace $\mathcal{K}$.*

*Proof.* Applying Lemma A.14 with $u = \Delta_{\text{adv}}$ and $v = \Delta$ and taking expectations yields

$$G \leq \mathbb{E}[|\langle \nabla \mathcal{L}_{\text{attack}}(\theta + \Delta),\, Z \rangle|] + \frac{L}{2}\mathbb{E}\big[\|Z\|_2^2\big].$$

By Cauchy–Schwarz,

$$\mathbb{E}[|\langle \nabla \mathcal{L}_{\text{attack}}(\theta + \Delta),\, Z \rangle|] \leq \mathbb{E}[\|\nabla_{\mathcal{K}}\mathcal{L}_{\text{attack}}(\theta + \Delta)\|_2] \cdot \sqrt{\mathbb{E}[\|Z\|_2^2]}.$$

Substituting Lemma A.15 and noting that $Z$ is supported on $\mathcal{K}$ yields the result. $\square$

This result underscores that the "attack fidelity" is intrinsically linked to the cumulative variance of the chosen coordinates. By selectively embedding backdoors into the subspace $\mathcal{K}$ where $\sum_{i \in \mathcal{K}} S_i$ is minimized, CTO-VPA effectively suppresses the aggregation noise, ensuring the global model's behavior closely aligns with the attacker's local optimization.

## B. Additional Evaluation

In this section, we provide additional experiments to validate the robustness of CTO-VPA and its subspace selection strategy. We analyze the sensitivity to the poisoning rate, malicious-client ratio, and top-$k$ subspace ratio, and further evaluate direct pruning and the Lockdown (Huang et al., 2024) defense.

## B.1. Ablation on Attack Budgets

We evaluate whether CTO-VPA relies on a specific attack budget by varying the poisoning rate (PR) and the malicious-client ratio (MR). As shown in Table 4 and Table 5, CTO-VPA remains effective across all tested configurations. Under poisoning rates $\{0.125, 0.25, 0.5\}$, CTO-VPA consistently achieves high BA and the longest lifespan, indicating that its persistence does not require a large poisoned data fraction. Similarly, when varying the malicious-client ratio among $\{0.05, 0.1, 0.2\}$, CTO-VPA remains the most persistent method even with only $5\%$ malicious clients. These results demonstrate that the advantage of CTO-VPA does not depend on a single hand-picked setting such as $10\%$ malicious clients or $25\%$ poisoning rate, but remains stable across different threat budgets.

*Table 4.* Ablation on poisoning rate (PR).

| Method | PR=0.125 | PR=0.25 | PR=0.5 | Lifespan $\lambda$ |
|---|---|---|---|---|
| CBA | 8.69 | 90.02 | 91.88 | 0/7/17 |
| DBA | 10.82 | 81.02 | 96.00 | 0/16/44 |
| Neurotoxin | 75.98 | 90.02 | 84.88 | 4/4/5 |
| FCBA | 89.78 | 90.89 | 99.08 | 5/17/15 |
| PFL | 57.49 | 95.48 | 96.80 | 13/35/40 |
| **CTO-VPA** | **99.90** | **100.00** | **100.00** | **92/255/290** |

*Table 5.* Ablation on malicious-client ratio (MR).

| Method | MR=0.05 | MR=0.1 | MR=0.2 | Lifespan $\lambda$ |
|---|---|---|---|---|
| CBA | 64.30 | 90.02 | 86.80 | 4/7/27 |
| DBA | 74.12 | 81.02 | 88.87 | 8/16/26 |
| Neurotoxin | 75.98 | 90.02 | 100.00 | 4/4/9 |
| FCBA | 87.30 | 90.89 | 85.69 | 9/17/15 |
| PFL | 97.33 | 95.48 | 96.96 | 40/35/42 |
| **CTO-VPA** | **100.00** | **100.00** | **100.00** | **71/255/272** |

## B.2. Sensitivity Analysis on the Top-$k$ Parameter Ratio

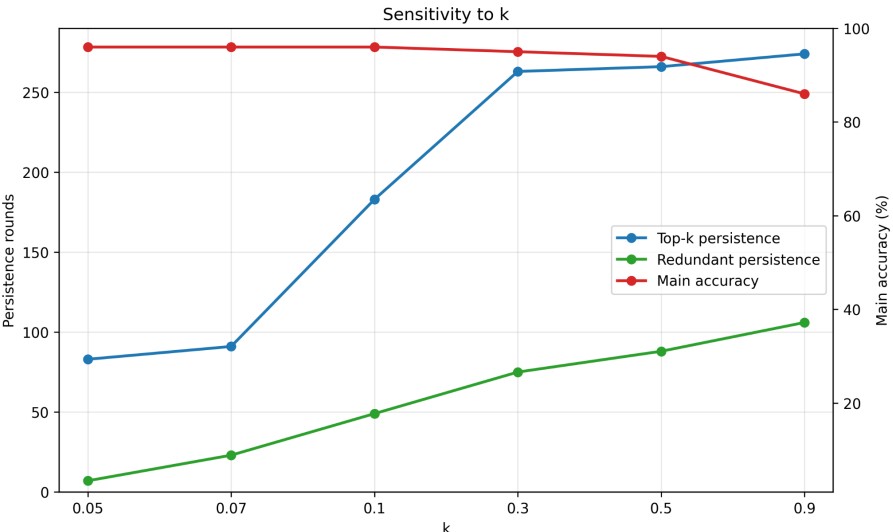

*Figure 6.* Sensitivity analysis on the top-$k$ parameter ratio.

We analyze the sensitivity of CTO-VPA to the top-$k$ subspace parameter ratio and compare it with redundant parameter (Neurotoxin). As shown in Figure 6, the persistence of the CTO-VPA-selected top-$k$ subspace increases as $k$ becomes larger, while the persistence of the redundant-parameter baseline remains substantially lower across all settings. The clear gap indicates that the proposed influence-score-based selection captures a functionally relevant parameter subset rather than an arbitrary or redundant one. Meanwhile, the main task accuracy remains high and nearly unchanged over a broad range of $k$, with only a mild decline when $k$ becomes very large. These results suggest that CTO-VPA is reasonably robust to the choice of $k$ and does not rely on a narrowly tuned parameter ratio.

## B.3. Pruning-based Analysis

To validate the difference and superiority of CTO-VPA's subspace selection strategy over the redundant-parameter selection commonly used in prior work (Zhang et al., 2022; Shi et al., 2024; Wu et al., 2021; Huang et al., 2025), we conduct a

single-shot pruning experiment under the same pruning budget. Specifically, we prune $10\%$ of parameters from three different sets: the CTO-VPA-selected top-$k$ subspace, a randomly selected parameter subset with the same budget, and the bottom-$k$ parameters with the lowest influence scores. The results are reported in Table 6. B-MA and B-ASR denote the main task accuracy and attack success rate before pruning, while A-MA and A-ASR denote the corresponding results after pruning. MA Ret. and ASR Ret. denote the retention rates of the main task and the backdoor task, respectively.

*Table 6.* Pruning analysis under the single-shot pruning.

| Pruning Set | B-MA | B-ASR | A-MA | A-ASR | MA Drop | ASR Drop | MA Ret. | ASR Ret. |
|---|---|---|---|---|---|---|---|---|
| Top-$k$ subspace | 89.45 | 100.00 | 18.83 | 0.36 | 70.62 | 99.64 | 21.05 | 0.36 |
| Random parameters | 89.37 | 95.28 | 68.78 | 90.48 | 20.59 | 4.80 | 76.96 | 94.96 |
| Bottom-$k$ parameters | 89.84 | 99.92 | 88.02 | 99.52 | 1.82 | 0.40 | 97.97 | 99.60 |

As shown in Table 6, pruning the top subspace selected by CTO-VPA $k$ reduces the ASR from $100.00\%$ to $0.36\%$, indicating that this subspace indeed carries the functionality of the implanted backdoor. However, this pruning also causes a severe decrease in MA from $89.45\%$ to $18.83\%$. In contrast, pruning random parameters only slightly reduces ASR from $95.28\%$ to $90.48\%$, while retaining $76.96\%$ of the clean accuracy. Pruning the bottom-$k$ parameters has an even smaller effect, with MA changing from $89.84\%$ to $88.02\%$ and ASR remaining at $99.52\%$.

These results indicate that the CTO-VPA-selected coordinates are not arbitrary or redundant. Instead, they encode persistent backdoor behavior while also substantially contributing to the clean task. This observation explains why directly pruning the selected subspace can suppress the backdoor but would simultaneously cause unacceptable degradation in benign model utility. Notably, although removing the decision-critical and optimization-stable subspace can eliminate the backdoor effect in this controlled analysis, such pruning is difficult to apply in practice. The defender does not have direct knowledge of the attacker-selected subspace and therefore cannot precisely identify the same coordinates. Moreover, aggressively pruning this subspace would substantially damage the main task performance.

### B.4. Evaluation under Lockdown

We further evaluate CTO-VPA under Lockdown (Huang et al., 2024), which defenses backdoor attacks based on isolated subspace training. Lockdown is closely related to the pruning concern because it attempts to mitigate backdoor effects by isolating and controlling the parameter subspaces involved in client training. Therefore, it provides a strong test for whether CTO-VPA can remain effective when the defense explicitly considers subspace-level backdoor behavior.

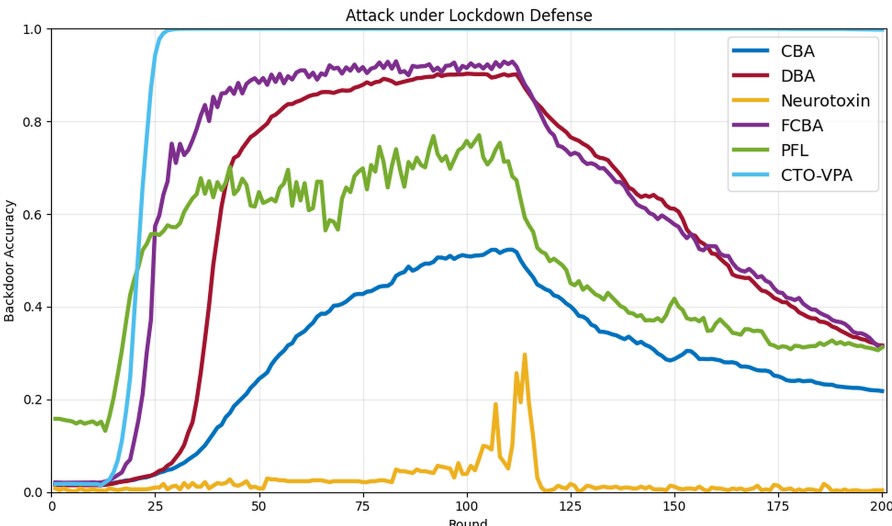

*Figure 7.* Backdoor accuracy under Lockdown defense.

Figure 7 reports the backdoor accuracy under Lockdown. CTO-VPA quickly reaches near-perfect backdoor accuracy and maintains a stable curve throughout the training process. In comparison, most baseline attacks exhibit substantially weaker

persistence. DBA and FCBA can reach relatively high BA during the active attack stage, but both exhibit clear decay once the defense continues to operate.

The superior persistence of CTO-VPA under Lockdown further supports that CTO-VPA does not rely on arbitrary large perturbations or dispensable redundant neurons, its backdoor behavior is less likely to be removed by simply isolating or pruning suspicious subspaces. Instead, the backdoor is embedded into coordinates that are both optimization-stable and decision-critical. As a result, pruning or suppressing these coordinates creates a difficult trade-off for the defender: removing the backdoor may also significantly damage the clean-task utility of the global model.

## C. Discussion

**Implications for Defense Design.** The results of this work suggest that defending against persistent backdoor attacks in federated learning requires going beyond per-round anomaly detection or magnitude suppression. While existing defenses mainly target large or statistically deviant updates, our findings show that backdoors embedded in optimization-stable and decision-critical subspaces can remain effective even when updates are small and distributionally benign. This exposes a limitation of defenses that operate locally in time or space. More effective protection may require reasoning about long-horizon optimization dynamics, tracking parameter stability across rounds, or identifying subspaces where benign and malicious objectives become inseparable. From this view, persistence emerges as a system-level property of aggregation and representation learning rather than a consequence of unusually strong attacks.

**Broader Perspective on Optimization Stability.** Although our study focuses on backdoor attacks, the optimization-based perspective developed here may have broader implications for understanding robustness in federated learning. Federated aggregation inherently amplifies parameter directions with consistent gradient contributions while suppressing unstable ones. This bias can unintentionally favor malicious behaviors that align with stable directions, but it may also explain other phenomena such as the persistence of spurious correlations or systematic biases in distributed models. More generally, our results suggest that security and robustness in federated learning are tightly coupled with representation learning and optimization geometry. Investigating how stability-driven subspaces emerge and evolve under different data distributions and training protocols may offer insights beyond the backdoor setting.

