# OpenReview forum: "Coupled Trigger Optimization and Vulnerable Parameter Alignment for Persistent Backdoor Attacks on Federated Learning"
_ICML.cc/2026/Conference — ICML 2026 regular_

### Official Review · Reviewer_NVU4 · 2026-03-08

**Soundness:** 3
**Presentation:** 2
**Significance:** 3
**Originality:** 3
**Overall Recommendation:** 5
**Confidence:** 3

**Summary:**

The paper proposes FL backdoor attack by constraining updates to the coupled subspace of optimization-stable directions and preserving benign performance.

**Compliance With Llm Reviewing Policy:**

Affirmed.

**Final Justification:**

The authors' additional empirical evidences partly addressed my original concerns.

**Key Questions For Authors:**

None

**Strengths And Weaknesses:**

# Strengths
* The paper correctly identifies that most FL backdoor attacks treat trigger design and parameter manipulation independently. Framing persistence as an optimization-dynamics problem is an interesting conceptual contribution.

# Weaknesses
* The paper claims 10% malicious clients and 25% poisoning rate for "all kinds of backdoor," but different attacks have very different design assumptions. The paper seems to miss that part.
* Three datasets (two of which are small-scale) and two architectures is borderline . There is no evaluation on larger-scale or more realistic FL scenarios (e.g., non-IID data distributions, varying participation rates, different numbers of clients). The paper mentions cross-device FL but tests with what seems to be a small number of clients in simulation.

---

> ### Author Rebuttal · Authors · 2026-03-30
>
> 1.We agree that different backdoor attacks may have different preferred operating regimes and design assumptions. To address this concern, we further added ablations on both the poisoning rate (PR) and the malicious-client rate (MR) in Table below (whole results in https://anonymous.4open.science/r/rebuttal_image-258F/malicious_rate.png  and https://anonymous.4open.science/r/rebuttal_image-258F/poisoning_rate.png ). The results show that the main conclusion is stable across different threat budgets. Specifically, under poisoning rates 0.125 0.25 0.5 and malicious-client fractions 0.05 0.1 0.2, our method consistently maintains substantially stronger backdoor persistence than the compared baselines, while keeping the main-task accuracy at a similar level. This suggests that the advantage of our method does not rely on a single hand-picked setting such as 10% malicious clients or 25% poisoning rate.
> Moreover, our method remains the most persistent curve in almost all settings, whereas most baselines decay quickly once the attack stops, especially under lower poisoning rates or fewer malicious clients.
> |Method|PR=0.125 |PR=0.25|PR=0.5| Lifespan|
> |---|---:|---:|---:|---:|
> |CBA| 8.69 | 90.02 | 91.88 | 0/7/17 |
> | DBA | 10.82 | 81.02 | 96.00 | 0/16/44 |
> | Neurotoxin | 75.98 | 90.02 | 84.88 | 4/4/5 |
> | FCBA | 89.78 | 90.89 | 99.08 | 5/17/15 |
> | PFL | 57.49 | 95.48 | 96.80 | 13/35/40 |
> | CTO-VPA | 99.90 | 100.00 | 100.00 | 92/255/290 |
>
> | Method | MR=0.05 | MR=0.1 | MR=0.2 | Lifespan |
> |---|---:|---:|---:|---:|
> | CBA | 64.30 | 90.02 | 86.80 | 4/7/27 |
> | DBA | 74.12 | 81.02 | 88.87 | 8/16/26 |
> | Neurotoxin | 75.98 | 90.02 | 100.00 | 4/4/9 |
> | FCBA | 87.30 | 90.89 | 85.69 | 9/17/15 |
> | PFL | 97.33 | 95.48 | 96.96 | 40/35/42 |
> | CTO-VPA | 100.00 | 100.00 | 100.00 | 71/255/272 |
>
> 2.In our experiments we also validate the performance on Tiny-ImageNet, which is substantially more challenging, and further validate the attack with a larger architecture, Astroformer (272M), to test the method in a more complex model setting. These results were intended to go beyond lightweight benchmarks and show that the attack remains effective on more demanding data/model combinations.
>
> We agree that realistic FL evaluation should include heterogeneous data distributions. To address this, we further evaluated the method under Dirichlet-partitioned non-IID settings with different heterogeneity levels in  https://anonymous.4open.science/r/rebuttal_image-258F/noniid_alpha.png . The results show that our method remains effective across all tested non-IID configurations. In particular, under the more extreme heterogeneity setting ($\alpha = 0.1$), our method shows a clear persistence advantage over the baselines. Under the moderate non-IID setting ($\alpha=0.5$), our method is comparable to PFL, which is expected because PFL is a personalized backdoor attack and is naturally better aligned with non-IID scenarios where client distributions are distinguishable. Even in this case, our method remains competitive while maintaining strong persistence. These results suggest that the proposed attack is not restricted to IID toy settings, and remains effective under more realistic heterogeneous client distributions.
>
> | Method | $\alpha=0.1$ | $\alpha=0.5$ |$\alpha=0.9$ | Lifespan |
> |---|---:|---:|---:|---:|
> | CBA | 59.43 | 83.28 | 54.92 | 2/11/2 |
> | DBA | 87.63 | 95.98 | 87.40 | 13/26/4 |
> | Neurotoxin | 64.65 | 100.00 | 100.00 | 5/8/11 |
> | FCBA | 99.98 | 94.99 | 92.81 | 22/80/16 |
> | PFL | 76.38 |100.00 | 94.60 | 8/200/20 |
> | CTO-VPA | 92.31 | 98.87 | 99.27 | 71/173/65 |

---

> > ### Author Rebuttal · Reviewer_NVU4 · 2026-04-03
> >
> > Thanks for the authors' efforts to address my previous concerns about having better empirical bounds across more attacks. I will raise my score.

---

> > > ### Author Response · Authors · 2026-04-06
> > >
> > > Thank you for your recognition of our efforts and for your positive reassessment of the paper.

---

### Official Review · Reviewer_fMZA · 2026-03-10

**Soundness:** 3
**Presentation:** 2
**Significance:** 2
**Originality:** 3
**Overall Recommendation:** 4
**Confidence:** 5

**Summary:**

The paper studies persistent backdoor attacks in federated learning from an optimization perspective. The main intuition is that backdoor effects fade over time because malicious updates are not well aligned with the parameter directions that are more likely to survive repeated federated aggregation. To mitigate this issue, the paper proposes CTO-VPA, which aims to jointly optimize the trigger and constrain the attack to a parameter subspace that is both optimization-stable and relevant to the target decision.

**Compliance With Llm Reviewing Policy:**

Affirmed.

**Final Justification:**

The rebuttal partially addressed my concerns and improved the paper’s clarity, but some issues remain. Overall, I slightly increase my score while maintaining a positive assessment.

**Key Questions For Authors:**

1. How exactly is the controlled scaling operation implemented?
2. Please clarify the full experimental setup, including the FL configuration, data partition strategy, attack injection round, and the specific privacy budget used in the DP defense.
3. Why were the defense experiments evaluated over 50 rounds only? Would it be more informative to present the full persistence curves under defenses, similar to Figure 2?
4. Can the authors add stronger recent baselines and more comprehensive ablations, especially under different malicious client ratios and poisoning ratios?

**Limitations:**

The limitations are largely consistent with the weaknesses discussed above, particularly regarding method clarity, experimental completeness, and empirical rigor.

**Strengths And Weaknesses:**

Strengths:
1. The paper addresses persistence in federated backdoor attacks, which is an important and practically relevant problem.
2. The motivation is easy to understand, and the paper looks beyond standard attack success rate by explicitly evaluating how long the attack remains effective.

Weaknesses:
1. The paper states that the selected parameters are updated with controlled scaling, but the main text does not provide the implementation details of this operation.
2. One of the claimed contributions is the coupling design, but in the current version it appears to be mainly realized through a heuristic strategy, without sufficient theoretical support.
3. Many important experimental details are missing, such as the federated learning configuration, the data partition protocol, the attack starting round, and the privacy budget setting in the DP-based defense.
4. The baseline comparison is not sufficiently strong. Some recent related methods are missing, for example: Infighting in the Dark: Multi-Label Backdoor Attack in Federated Learning (CVPR 2025).
5. In the defense experiments, the persistence evaluation is reported over 50 rounds only. It is unclear why this evaluation horizon is chosen, instead of presenting continuous multi-round results as in Figure 2.
6. The paper does not report statistical significance.
7. The paper lacks analysis on several key factors, such as different malicious client ratios and different poisoning ratios.

---

> ### Author Rebuttal · Authors · 2026-03-30
>
> 1.The scaling step in our implementation follows the standard setting adopted in How To Backdoor Federated Learning. The selected backdoor parameters in subspace are scaled by $\frac{n}{\eta}$, where n is the client participation ratio and $\eta$ is the global learning rate. The scaled parameters are then projected back into the prescribed bounded interval, so that the final malicious update stays within a normal parameter range. The current manuscript states the motivation of this operation but does not describe the concrete implementation in sufficient detail. We will revise the main text to explicitly present this scaling-and-projection step and cite the corresponding reference.
>
> 2.The experimental setup: total 100 clients, in each round 10% of clients are sampled to participate. Among the sampled clients, 10% are malicious.  The top-k ratio is 30%, poisoned data rate is 25%, the sharpness parameter is alpha=1.5, the window length is N=30. Consistent with the common practice in backdoor attacks, the attack is injected near convergence that the global model is first pre-trained for 200 rounds on GTSRB and 300 rounds on CIFAR-10 and Tiny-ImageNet, and the malicious update is introduced afterward. Under the DP defense, the privacy budget is set to $\epsilon=5$ and $\delta=10^-5$.
>
> 3.We reported the 50-round persistence summary mainly for space efficiency and readability. The full persistence curves are provided at https://anonymous.4open.science/r/rebuttal_image-258F/asr_under_defense.png . In our setup, the attack stops at round 113, and all subsequent rounds are clean training rounds with the corresponding defense applied, during which we track the residual backdoor behavior.
>
> 4.We have already discussed Infighting in the Dark: Multi-Label Backdoor Attack in Federated Learning in the related work. It was not included in the original experiments because its primary focus differs from ours, that work studies the non-cooperative multi-label backdoor attack, while our paper focuses on the persistence of the implanted backdoor, especially its survival under subsequent aggregation and defenses.
> To make the baseline comparison stronger, we have now added it under the same experimental setting, and include its multi-shot-2 performance in the figure and table below. The figure https://anonymous.4open.science/r/rebuttal_image-258F/mirage_persistence.png  records the attack persistence after continuous attack stops. As shown, Mirage can achieve very high attack success during attack, but its post-attack persistence is unstable because it relies on three non-colluding malicious clients with different targets, leading to uneven survival across clients. In contrast, CTO-VPA exhibits more stable and consistently stronger persistence after the attack stops.
> As shown in Table, Mirage achieves nearly perfect ASR for all three clients at the end of attack (mean ASR =99.91%), indicating that it is indeed a strong recent baseline. However, after the attack stops, its persistence becomes uneven across clients. This suggests that Mirage is highly effective in the non-cooperative multi-label setting, but its long-horizon post-attack persistence is less stable across targets.
> | Method | ASR at end of attack | ASR after 100 rounds attack stop | Average ASR over 100 post-attack rounds |
> |---|---:|---:|---:|
> | Mirage Client 1 | 99.94 | 94.42 | 97.96 |
> | Mirage Client 2 | 99.93 | 99.29 | 99.66 |
> | Mirage Client 3 | 99.86 | 28.62 | 60.36 |
> | Mirage Mean | 99.91 | 74.11 | 85.99 |
> | CTO-VPA | 100.00 | 96.20 | 99.16 |
>
> 5.For the ablation on the malicious client fraction and the poisoned data rate, please refer to our response to NVU4. Comment 1.
>
> 6.In the appendix, we provide theoretical analyses of the proposed optimization view from multiple perspectives, including the convergence analysis of the coupled projected optimization procedure in Appendix A.1 and the tight aggregation-gap bound showing how attack fidelity depends on the cumulative variance of the selected subspace coordinates. These analyses are built directly on Eqs. (13)–(16), where trigger optimization and parameter manipulation are jointly formulated as a constrained projected optimization problem rather than two independent steps.
> The attack gradient is restricted to the selected stable subspace K, while clean optimization is preserved on the complementary coordinates. Therefore, the trigger-induced malicious signal is forced to propagate through aggregation-stable directions. Combined with the appendix aggregation-gap analysis, this implies that coupling reduces the distortion of the malicious update after aggregation compared with a decoupled design that allows attack energy to leak into unstable coordinates. In this sense, the coupling design has a clear optimization-based justification for improving persistence, rather than being a purely heuristic choice.

---

> > ### Author Rebuttal · Reviewer_fMZA · 2026-04-03
> >
> > The authors have partially addressed my concerns. However, some issues remain. I slightly increase my score.

---

> > > ### Author Response · Authors · 2026-04-06
> > >
> > > Thank you for acknowledging the improvements we made in response to your concerns.

---

### Official Review · Reviewer_QGMn · 2026-03-13

**Soundness:** 3
**Presentation:** 3
**Significance:** 3
**Originality:** 4
**Overall Recommendation:** 4
**Confidence:** 4

**Summary:**

This paper studies the persistence of backdoor attacks in federated learning (FL) from an optimization perspective. The authors argue that long-term effective backdoors require alignment between trigger-induced representations and parameter directions that remain stable under aggregation. Based on this insight, the paper proposes CTO-VPA (Coupled Trigger Optimization and Vulnerable Parameter Alignment), which jointly optimizes trigger representations while constraining updates to a decision-critical, optimization-stable subspace. Experimental results on CIFAR-10, GTSRB, and Tiny-ImageNet show that CTO-VPA achieves higher backdoor persistence and robustness compared to several state-of-the-art attacks, without significantly degrading main task accuracy. Ablation studies further support the importance of trigger–parameter coupling and subspace selection.

**Compliance With Llm Reviewing Policy:**

Affirmed.

**Final Justification:**

The rebuttal adequately addressed most of my concerns. To further improve readability, I suggest the authors include an algorithm. I will raise my score to 4 (weak accept).

**Key Questions For Authors:**

See Weaknesses.

**Limitations:**

yes

**Strengths And Weaknesses:**

Strengths:
Novel perspective: The paper analyzes backdoor persistence through optimization dynamics, highlighting the importance of embedding backdoor signals in stable parameter directions.

Clear method design: CTO-VPA explicitly couples trigger optimization with constrained updates in a stable, decision-critical subspace, providing a mechanism-level explanation for persistent attacks.

Extensive experiments: The evaluation covers multiple datasets, attack schedules (single-shot and multi-shot), and defensive mechanisms (Norm-Clipping, Trimmed-Mean, DP, FLAME), demonstrating both persistence and stealth.

Stealthy attack: CTO-VPA maintains main task accuracy while achieving long-lasting backdoor effects, outperforming baselines that either degrade accuracy or decay rapidly.


Weaknesses

Inconsistency.  Eq. (13) says parameters outside $\mathcal{K}$ are updated with the clean gradient, while Eq. (14) projects both attack and clean gradients onto $\mathcal{K}$.

Limited discussion of client data heterogeneity (non-IID): Federated learning often involves non-IID data distributions across clients, which can affect both optimization dynamics and backdoor persistence. The paper does not explicitly study or discuss how non-IID data impacts CTO-VPA’s effectiveness.

Sensitivity analyses are missing. The method depends on many choices: top-k% parameter selection, projection bound $\epsilon$, data poison rate, malicious client fraction, and the clean/attack trade-off $\lambda$.

Theoretical analysis not fully integrated in the main text: While the paper discusses optimization-based motivations, it lacks formal theoretical derivations or citations in the main body. If relevant analysis exists in the appendix, it should be referenced in the main text to improve clarity and readability.

Missing evaluation on recent FL backdoor defenses: While CTO-VPA is tested against Norm-Clipping, Trimmed-Mean, DP, and FLAME, it is not evaluated against some of the latest SOTA federated backdoor defenses (e.g.,Alignins[1]). Including these comparisons would strengthen the claims about robustness and persistence.

The ablation study is not sufficient. The paper needs a more systematic decomposition of trigger learning, parameter selection, projection, and scaling.

[1]Detecting Backdoor Attacks in Federated Learning via Direction Alignment  Inspection. CVPR 2025.

---

> ### Author Rebuttal · Authors · 2026-03-30
>
> 1.Eqs. (13) and (14) are intended to express the same update rule in coordinate-wise and compact forms, respectively. Specifically, the attack gradient is restricted to the selected subspace K, while the clean gradient is restricted to its complementary subspace \bar{K}. Under this interpretation, Eq. (14) is the vectorized form of Eq. (13), rather than a different optimization rule. The ambiguity arises because the subscript indicating the complementary projection in Eq. (14) is visually small and therefore not sufficiently noticeable in the current manuscript. In addition, the text does not explicitly state that the clean gradient is applied on (\bar{K}), which makes this point easy to misread. We will revise the equation presentation and add explicit textual clarification that the clean gradient is restricted to (\bar{K}), so that the consistency between Eqs. (13) and (14) is clear and unambiguous.
>
> 2.For the Non-IID experiments, please refer to our response to NVU4. Comment 2.
>
> 3.We have added a sensitivity analysis on the top-k parameter ratio in https://anonymous.4open.science/r/rebuttal_image-258F/sensitive_k.png . The results show that, as k increases, the persistence of the selected top-k subspace improves consistently, whereas the persistence of redundant parameters (Neurotoxin-based model poisoning with our trigger-coupling training) remains substantially lower across all settings. This clear gap suggests that the proposed influence-score-based selection captures a functionally relevant parameter subset, rather than an arbitrary choice. Meanwhile, the main-task accuracy remains high and nearly unchanged over a broad range of k, with only a mild decline when k becomes very large. Overall, these results suggest that the method is reasonably robust to the choice of k and does not rely on a narrowly tuned selection ratio.
>
> 4.In the appendix, we provide theoretical analyses on the convergence-oriented interpretation of the coupled projected optimization, the coordinate-wise SNR analysis for backdoor persistence, and the aggregation-gap bound that links persistence to the variance accumulated on the selected subspace. These results were included to formalize the optimization-based motivation in the paper. We agree that the current main text does not reference these appendix analyses clearly enough. In the revised version, we will explicitly cite the corresponding appendix results in the main text.
>
> 5.We agree that Alignins reduces the effectiveness of our attack. Nevertheless, CTO-VPA still consistently outperforms all baselines. Although Alignins can suppress part of the malicious effect by filtering suspicious updates, CTO-VPA still preserves substantially stronger attack strength and persistence than prior methods. This demonstrates that our method is harder to neutralize than existing backdoor attacks, even when faced with a recent direction-inspection-based defense.
> | Method | Value |Lifespan|
> |---|---|---:|
> | CBA | 5.29 | 0 |
> | DBA | 5.29 | 0 |
> | Neurotoxin | 0.39 | 0 |
> | FCBA | 6.28 | 0 |
> | PFL | 25.41 | 0 |
> | CTO-VPA | 51.93 | 21 |
>
> 6.The ablation on the malicious client fraction and the poisoned data rate, please refer to our response to NVU4. Comment 1. For the parameter-selection experiment on the critical subspace, please refer to our response to OSAV, Comment 1.

---

> > ### Author Rebuttal · Reviewer_QGMn · 2026-04-03
> >
> > The rebuttal adequately addressed most of my concerns. To further improve readability, I suggest the authors include an algorithm. I will raise my score to 4 (weak accept).

---

> > > ### Author Response · Authors · 2026-04-06
> > >
> > > Thank you for your feedback and for recognizing the improvements in the revised manuscript. Following your helpful suggestion, we will include an algorithm in the revision to further improve readability and clarify the overall procedure, which we believe will further strengthen the quality of the manuscript.
> > >
> > > ### Algorithm 1: Training Procedure of CTO-VPA
> > >
> > > **Input:**
> > > $\theta^G,\ D_{cl},\ D_{bd},\ N, k$
> > >
> > > **Output:**
> > > $\Delta\theta,\ \tau,\ K$
> > >
> > > 1. Initialize $\tau=\{t_r,t_g,t_b\}$.
> > >
> > > 2. For each parameter $w_i$, compute
> > >    $\Delta w_i^{(t)} = w_i^{(t)} - w_i^{(t-1)}$.
> > >
> > > 3. Over an $N$-round window, compute
> > > $$
> > > I_i = \frac{1}{N}\sum_{r=1}^{N}\Delta w_i^{(r)},
> > > \qquad
> > > S_i = \frac{1}{N-1}\sum_{r=1}^{N}\left(\Delta w_i^{(r)}-I_i\right)^2.
> > > $$
> > >
> > > 4. $$F_i=\left(\frac{|I_i|}{\sqrt{S_i}+\kappa}\right)^\alpha.$$
> > >
> > > 5. Select $$ K=TopK (\\{F_i\\}_{i=1}^P,k\\%).$$
> > >
> > > 6. $$
> > > L(\theta,\tau)=L_{attack}(\theta,\tau)+\lambda L_{clean}(\theta).
> > > $$
> > >
> > > 7. Update trigger $$\tau \leftarrow \tau-\eta_\tau \nabla_\tau L_{attack}(\theta,\tau).$$
> > >
> > > 8. Update parameters
> > > $$
> > > \Delta w_i = -\eta \nabla_{w_i}L_{attack}(\theta,\tau), \quad i\in K
> > > $$
> > >
> > > $$
> > > \Delta w_i = -\eta \nabla_{w_i}L_{clean}(\theta), \quad i\notin K
> > > $$
> > >
> > > $$
> > > g(\theta,\tau)=P_K\nabla_\theta L_{attack}(\theta,\tau)+P_{\bar K}\nabla_\theta L_{clean}(\theta).
> > > $$
> > >
> > > 9. Project selected updates
> > > $$
> > > w_i^{new}=Proj_{\|\cdot\|_\infty\le \epsilon}\left(w_i^{old}+\Delta w_i\right), \quad i\in K.
> > > $$
> > >
> > > 10. Return
> > > $$
> > > \Delta\theta=\theta^{new}-\theta^{old}.
> > > $$

---

### Official Review · Reviewer_osav · 2026-03-14

**Soundness:** 2
**Presentation:** 3
**Significance:** 2
**Originality:** 2
**Overall Recommendation:** 3
**Confidence:** 4

**Summary:**

The paper proposes CTO-VPA, which constrains backdoor updates to a coupled subspace to improve attack persistence. The method selects the coupled subspace based on optimization stability and backdoor influence, so that backdoor behaviors can be embedded into optimization-stable directions while preserving benign performance.

**Compliance With Llm Reviewing Policy:**

Affirmed.

**Final Justification:**

Authors partially solved my concerns thus raise the score.

**Key Questions For Authors:**

C1. Questions on the motivation of CTO-VPA, which aims to implant backdoor behavior into a compact parameter subspace that is both decision-critical for activation and inherently stable under federated aggregation (lines 62–65, 144–147). The paper does not discuss whether this “stable subspace” overlaps with redundant neurons studied in prior work [1][2][3], nor whether the proposed attack would remain effective under pruning-based defenses [2][3][4]. This leaves a central ambiguity: if the selected coordinates are effectively redundant, the method may be vulnerable to pruning; if not, the paper should clarify how they differ from redundancy-based parameter subsets.

C2. According to C1, I am inclined to think that the stable subspace highly overlaps with redundant neurons [1][2][3][5], especially given the computation of the backdoor influence score in Eq. (12). Therefore, the method appears similar to a combination of Neurotoxin and 3DFed, rather than a fundamentally new attack framework.

C3. Threat model is unclear. The computation of the backdoor influence score appears to require the attackers to participate in N consecutive rounds in order to estimate the update statistics in Eqs. (10) and (11). However, this assumption seems impractical. Moreover,  the paper itself provides a sensitivity analysis on the window length NNN, and the results show that the method is sensitive to this choice.

C4. Please provide complete default hyperparameter settings. For example, the paper does not clearly specify the default value of parameter k. Other key hyperparameters (e.g., FL communication round, sample fraction) should also be reported more clearly for reproducibility.

C5. Please provide more evaluation under different malicious client ratios (especially below 10%) and different Non-IID degrees.

[1] Li H, Ye Q, Hu H, et al. 3dfed: Adaptive and extensible framework for covert backdoor attack in federated learning

[2] Huang T, Hu S, Chow K H, et al. Lockdown: backdoor defense for federated learning with isolated subspace training

[3] Han S, Park S, Wu F, et al. Towards attack-tolerant federated learning via critical parameter analysis

[4] Yan P, Wang H, Song T, et al. SkyMask: Attack-agnostic robust federated learning with fine-grained learnable masks

**Limitations:**

yes

**Strengths And Weaknesses:**

Strength:

++ The organization and writing of the paper are clear and easy to follow.

++ The paper studies backdoor persistence from an optimization perspective, which is a potentially interesting angle



Weakness:

-- The novelty is limited, especially in relation to prior parameter-subspace / neuron-selection-based attack methods; see C2.

-- There are several methodological concerns regarding the motivation, formulation, and justification of the proposed stable subspace and influence score; see C1, C2, and C3.

-- The evaluation is still limited in several important aspects, including default hyperparameter reporting, attack settings, and stronger validation of the claimed mechanism; see C4, C5.

---

> ### Author Rebuttal · Authors · 2026-03-30
>
> 1.Our optimization-stable subspace is different from the redundant neurons in prior works. Redundant neurons are defined as parameters that can be removed with little impact on the clean task. In contrast, CTO-VPA selects a decision-critical and aggregation-preserved parameter subset using the backdoor influence score $F_i$, where a larger score is assigned to coordinates with a stronger coherent update signal across rounds and lower aggregation-induced variation. As defined in Sec. 4.2, this criterion favors parameters that are consistently preserved by federated optimization dynamics, rather than dormant or dispensable ones.
> Therefore, our selected subspace should not be viewed as merely a redundancy-based subset. If the CTO-VPA-selected coordinates were merely redundant, pruning them under a modest budget would be expected to have limited effect on either the clean model or the backdoor. The results under the same one-shot pruning budget (10% pruning, top-k=10%) are shown below, pruning the top-k subspace reduces ASR from 100.00% to 0.36% and MA from 89.45% to 18.83%. Pruning a random subset of the same size only changes ASR from 95.28% to 90.48%, while pruning the non-selected bottom-k parameters leaves the attack almost unchanged (99.92% to 99.52%). This gap indicates that the coordinates selected by CTO-VPA are not arbitrary or redundant, they encode the persistent backdoor behavior and are also important to maintaining model functionality.
>
> | Pruning set | Before-pruning MA | Before-pruning ASR | After-pruning MA | After-pruning ASR | MA drop | ASR drop | MA retention | ASR retention |
> |---|---:|---:|---:|---:|---:|---:|---:|---:|
> |top-k subspace | 89.45 | 100.00 | 18.83 | 0.36 | 70.62 | 99.64 | 21.05% | 0.36% |
> |Random parameters| 89.37 | 95.28 | 68.78 | 90.48 | 20.59 | 4.80 | 76.96% | 94.96% |
> |bottom-k parameters| 89.84 | 99.92 | 88.02 | 99.52 | 1.82 | 0.40 | 97.97% | 99.60% |
>
> These results support our central claim: CTO-VPA identifies a backdoor-critical stable subspace, rather than a set of dispensable redundant parameters. This also explains why pruning-based defenses are not straightforward in this setting. In practice, the defender does not know the attacker-selected subspace a priori, and our results further suggest that forcibly pruning this subspace would severely damage the clean-task utility.
>
> Please refer to our response to Reviewer QGMn, Comment 3, where we compare the sensitivity of redundant neurons and the top-k parameters. In all cases, the persistence of CTO-VPA remains consistently stronger than that of the redundant-neuron baseline.
>
> 2.CTO-VPA does not merely manipulate backdoor parameters. It jointly optimizes (i) a learnable fine-grained trigger, (ii) a backdoor-critical and optimization-stable parameter subspace, and (iii) a coupled training strategy that explicitly binds the trigger to the selected subspace. Thus, the core novelty lies in the trigger–subspace coupling, rather than in parameter selection alone.
> More specifically, the trigger is not a fixed handcrafted pattern, but a learnable channel-wise mask that is initialized from zero and gradually optimized during backdoor training. This enables the trigger to adapt to FL dynamics and activate the backdoor more stably. Meanwhile, the selected subspace constrains where the backdoor update is embedded, so trigger optimization and parameter manipulation are learned in a tightly coupled manner. Its selected coordinates are not redundancy-based, but form a compact subspace that jointly captures backdoor influence and aggregation stability. More importantly, subspace selection is only one component, the key lies in the bidirectional coupling between the trigger and the backdoor-critical subspace, which strengthens backdoor persistence.
>
> 3.The N-round requirement does not mean that the attacker must be selected in N consecutive rounds. It only requires access to, or a practical estimate of, the model trajectory over an N-round window.
> This can arise in two realistic cases. i) Observable global trajectory: in many FL settings, the released global models can be passively collected over multiple rounds, so the attacker can compute the required statistics without being selected every round. ii) Restricted observability: in more centralized settings, the attacker may instead use short-horizon local look-ahead simulation on its own data to obtain a practical surrogate of the dominant optimization direction.
> The performance variation of N is acceptable, that does not undermine practicality even under the least favorable tested settings ($N=5$ or $50$) in the single-shot, the attack still remains above 50% for over 150 rounds, which is longer than all baselines (all below 50 rounds).
>
> 4.top-k=30%, FL pre-trains 300 global rounds for CIFAR-10 and Tiny-Imagenet 200 rounds for GTSRB, clients_num=100, sample fraction=10%, malicious_client=10%, poisoned_rate=25%.
>
> 5.Non-IID experiments please refer to our response to NVU4. Comment 2.

---

> > ### Author Rebuttal · Reviewer_osav · 2026-04-03
> >
> > Thank you to the authors for the detailed rebuttal, which has addressed a large part of my concerns.
> >
> > **Regarding 1 and 2**, thank you for the clarification. However, I still believe it is necessary to evaluate the proposed method under more recent prune-based defenses, since I remain concerned about whether the attack can still remain effective when the selected subspace is subject to pruning. Moreover, the current defense coverage also appears somewhat outdated, and I believe a stronger evaluation on more recent defenses would substantially strengthen the paper.
> >
> > Therefore, I will slightly raise my score to 3.

---

> > > ### Author Response · Authors · 2026-04-06
> > >
> > > Thank you for acknowledging our clarification.
> > >
> > > To further address the concern regarding prune-based defenses, we have added experiments under the *Lockdown* defense, where pruning is applied in every round. In this setting, the multi-shot attack stops at round 113, and the defense continues thereafter. The results in https://anonymous.4open.science/r/rebuttal_image-258F/lockdown.png show that CTO-VPA not only achieves near-perfect backdoor accuracy during the attack stage, but also preserves the strongest post-attack persistence among all compared methods once the attack is removed. By contrast, the baselines exhibit substantially faster degradation under continuous pruning, and some quickly lose their backdoor effect almost entirely. CTO-VPA still preserves substantially stronger post-attack persistence than the baselines, indicating that its backdoor signal is embedded in parameter directions that remain effective even under *Lockdown*’s isolated-subspace training and continual pruning.

---

### Decision · Program_Chairs · 2026-04-30

**Decision:**

Accept (regular)

**Comment:**

This work studies the persistence of backdoors in distributed/federated learning  from an optimization perspective. The paper argues that for backdoors to remain effective over the long term, trigger-induced representations must align with parameter directions that remain stable during aggregation. Based on this premise, the authors introduce CTO-VPA (Coupled Trigger Optimization and Vulnerable Parameter Alignment), a method designed to optimize trigger representations while restricting parameter updates to a subspace that they identify as decision-critical and stable. The authors present experimental results on CIFAR-10, GTSRB, and Tiny-ImageNet, which indicate that CTO-VPA achieves greater backdoor persistence and robustness compared to existing attacks, without substantially reducing main task accuracy. Additionally, they cite ablation studies as supporting evidence for the importance of the proposed trigger–parameter coupling and subspace selection.

Except reviewer osav who maintains the necessity to evaluate the proposed method under more recent prune-based defenses, most reviewers proposed a positive assessment of the paper (or raised their score to a positive assessment).

While the clarity of the writing could be further improved, which I hope will be addressed in the camera ready, the paper constitutes a contribution worthy of being presented at ICML.